# GENERALIZED KERNEL THINNING

**Raaz Dwivedi**[1]**, Lester Mackey**[2]
[1] Department of Computer Science, Harvard University and Department of EECS, MIT
[2] Microsoft Research New England
raaz@mit.edu, lmackey@microsoft.com

## ABSTRACT

The kernel thinning (KT) algorithm of Dwivedi and Mackey (2021) compresses a probability distribution more effectively than independent sampling by targeting a reproducing kernel Hilbert space (RKHS) and leveraging a less smooth square-root kernel. Here we provide four improvements. First, we show that KT applied directly to the target RKHS yields tighter, dimension-free guarantees for any kernel, any distribution, and any fixed function in the RKHS. Second, we show that, for analytic kernels like Gaussian, inverse multiquadric, and sinc, target KT admits maximum mean discrepancy (MMD) guarantees comparable to or better than those of square-root KT without making explicit use of a square-root kernel. Third, we prove that KT with a fractional power kernel yields better-than-Monte-Carlo MMD guarantees for non-smooth kernels, like Laplace and Matérn, that do not have square-roots. Fourth, we establish that KT applied to a sum of the target and power kernels (a procedure we call KT+) simultaneously inherits the improved MMD guarantees of power KT and the tighter individual function guarantees of target KT. In our experiments with target KT and KT+, we witness significant improvements in integration error even in 100 dimensions and when compressing challenging differential equation posteriors.

## 1 INTRODUCTION

A core task in probabilistic inference is learning a compact representation of a probability distribution $\mathbb{P}$. This problem is usually solved by sampling points $x_1, \ldots, x_n$ independently from $\mathbb{P}$ or, if direct sampling is intractable, generating $n$ points from a Markov chain converging to $\mathbb{P}$. The benefit of these approaches is that they provide asymptotically exact sample estimates $\mathbb{P}_{\text{in}} f \triangleq \frac{1}{n} \sum_{i=1}^{n} f(x_i)$ for intractable expectations $\mathbb{P}f \triangleq \mathbb{E}_{X \sim \mathbb{P}}[f(X)]$. However, they also suffer from a serious drawback: the learned representations are unnecessarily large, requiring $n$ points to achieve $|\mathbb{P}f - \mathbb{P}_{\text{in}}f| = \Theta(n^{-\frac{1}{2}})$ integration error. These inefficient representations quickly become prohibitive for expensive downstream tasks and function evaluations: for example, in computational cardiology, each function evaluation $f(x_i)$ initiates a heart or tissue simulation that consumes 1000s of CPU hours (Niederer et al., 2011; Augustin et al., 2016; Strocchi et al., 2020).

To reduce the downstream computational burden, a standard practice is to *thin* the initial sample by discarding every $t$-th sample point (Owen, 2017). Unfortunately, standard thinning often results in a substantial loss of accuracy: for example, thinning an i.i.d. or fast-mixing Markov chain sample from $n$ points to $n^{\frac{1}{2}}$ points increases integration error from $\Theta(n^{-\frac{1}{2}})$ to $\Theta(n^{-\frac{1}{4}})$.

The recent *kernel thinning* (KT) algorithm of Dwivedi & Mackey (2021) addresses this issue by producing thinned coresets with better-than-i.i.d. integration error in a reproducing kernel Hilbert space (RKHS, Berlinet & Thomas-Agnan, 2011). Given a target kernel[1] $\mathbf{k}$ and a suitable sequence of input points $\mathcal{S}_{\text{in}} = (x_i)_{i=1}^{n}$ approximating $\mathbb{P}$, KT returns a subsequence $\mathcal{S}_{\text{out}}$ of $\sqrt{n}$ points with better-than-i.i.d. *maximum mean discrepancy* (MMD, Gretton et al., 2012),[2]

$$\text{MMD}_{\mathbf{k}}(\mathbb{P}, \mathbb{P}_{\text{out}}) \triangleq \sup_{\|f\|_{\mathbf{k}} \le 1} |\mathbb{P}f - \mathbb{P}_{\text{out}}f| \quad \text{for} \quad \mathbb{P}_{\text{out}} \triangleq \frac{1}{\sqrt{n}} \sum_{x \in \mathcal{S}_{\text{out}}} \boldsymbol{\delta}_x, \tag{1}$$

---

[1] A kernel $\mathbf{k}$ is any function that yields positive semi-definite matrices $(\mathbf{k}(z_i, z_j))_{i,j=1}^{l}$ for all inputs $(z_i)_{i=1}^{l}$.
[2] MMD is a metric for *characteristic* $\mathbf{k}$, like those in Tab. 1, and controls integration error for all bounded continuous $f$ when $\mathbf{k}$ *determines convergence*, like each $\mathbf{k}$ in Tab. 1 except SINC (Simon-Gabriel et al., 2020).

where $\|\cdot\|_{\mathbf{k}}$ denotes the norm for the RKHS $\mathcal{H}$ associated with $\mathbf{k}$. That is, the KT output admits $o(n^{-\frac{1}{4}})$ worst-case integration error across the unit ball of $\mathcal{H}$.

KT achieves its improvement with high probability using non-uniform randomness and a less smooth *square-root kernel* $\mathbf{k}_{\mathrm{rt}}$ satisfying

$$\mathbf{k}(x,y) = \int_{\mathbb{R}^d} \mathbf{k}_{\mathrm{rt}}(x,z)\mathbf{k}_{\mathrm{rt}}(z,y)dz. \tag{2}$$

When the input points are sampled i.i.d. or from a fast-mixing Markov chain on $\mathbb{R}^d$, Dwivedi & Mackey prove that the KT output has, with high probability, $\mathcal{O}_d(n^{-\frac{1}{2}}\sqrt{\log n})$-MMD$_{\mathbf{k}}$ error for $\mathbb{P}$ and $\mathbf{k}_{\mathrm{rt}}$ with bounded support, $\mathcal{O}_d(n^{-\frac{1}{2}}(\log^{d+1} n \log\log n)^{\frac{1}{2}})$-MMD$_{\mathbf{k}}$ error for $\mathbb{P}$ and $\mathbf{k}_{\mathrm{rt}}$ with light tails, and $\mathcal{O}_d(n^{-\frac{1}{2}+\frac{d}{2\rho}}\sqrt{\log n \log\log n})$-MMD$_{\mathbf{k}}$ error for $\mathbb{P}$ and $\mathbf{k}_{\mathrm{rt}}^2$ with $\rho > 2d$ moments. Meanwhile, an i.i.d. coreset of the same size suffers $\Omega(n^{-\frac{1}{4}})$ MMD$_{\mathbf{k}}$. We refer to the original KT algorithm as ROOT KT hereafter.

**Our contributions** In this work, we offer four improvements over the original KT algorithm. First, we show in Sec. 2.1 that a generalization of KT that uses only the target kernel $\mathbf{k}$ provides a tighter $\mathcal{O}(n^{-\frac{1}{2}}\sqrt{\log n})$ integration error guarantee for each function $f$ in the RKHS. This TARGET KT guarantee (a) applies to **any kernel k on any domain** (even kernels that do not admit a square-root and kernels defined on non-Euclidean spaces), (b) applies to **any target distribution** $\mathbb{P}$ (even heavy-tailed $\mathbb{P}$ not covered by ROOT KT guarantees), and (c) is **dimension-free**, eliminating the exponential dimension dependence and $(\log n)^{d/2}$ factors of prior ROOT KT guarantees.

Second, we prove in Sec. 2.2 that, for analytic kernels, like Gaussian, inverse multiquadric (IMQ), and sinc, TARGET KT admits MMD guarantees comparable to or better than those of Dwivedi & Mackey (2021) without making explicit use of a square-root kernel. Third, we establish in Sec. 3 that generalized KT with a fractional $\alpha$-power kernel $\mathbf{k}_\alpha$ yields improved MMD guarantees for kernels that do not admit a square-root, like Laplace and non-smooth Matérn. Fourth, we show in Sec. 3 that, remarkably, applying generalized KT to a sum of $\mathbf{k}$ and $\mathbf{k}_\alpha$—a procedure we call *kernel thinning+* (KT+)—simultaneously inherits the improved MMD of POWER KT and the dimension-free individual function guarantees of TARGET KT.

In Sec. 4, we use our new tools to generate substantially compressed representations of both i.i.d. samples in dimensions $d = 2$ through $100$ and Markov chain Monte Carlo samples targeting challenging differential equation posteriors. In line with our theory, we find that TARGET KT and KT+ significantly improve both single function integration error and MMD, even for kernels without fast-decaying square-roots.

| GAUSS$(\sigma)$ $\sigma > 0$ | LAPLACE$(\sigma)$ $\sigma > 0$ | MATÉRN$(\nu, \gamma)$ $\nu > \frac{d}{2}, \gamma > 0$ | IMQ$(\nu, \gamma)$ $\nu > 0, \gamma > 0$ | SINC$(\theta)$ $\theta \neq 0$ | B-SPLINE$(2\beta+1, \gamma)$ $\beta \in \mathbb{N}$ |
|---|---|---|---|---|---|
| $\exp\left(-\frac{\|z\|_2^2}{2\sigma^2}\right)$ | $\exp\left(-\frac{\|z\|_2}{\sigma}\right)$ | $c_{\nu-\frac{d}{2}}(\gamma\|z\|_2)^{\nu-\frac{d}{2}}$ $\cdot K_{\nu-\frac{d}{2}}(\gamma\|z\|_2)$ | $\frac{1}{(1+\|z\|_2^2/\gamma^2)^\nu}$ | $\prod_{j=1}^d \frac{\sin(\theta z_j)}{\theta z_j}$ | $\mathfrak{B}_{2\beta+2}^{-d} \prod_{j=1}^d h_\beta(\gamma z_j)$ |

**Table 1: Common kernels k$(x,y)$ on $\mathbb{R}^d$ with $z = x - y$.** In each case, $\|\mathbf{k}\|_\infty = 1$. Here, $c_a \triangleq \frac{2^{1-a}}{\Gamma(a)}$, $K_a$ is the modified Bessel function of the third kind of order $a$ (Wendland, 2004, Def. 5.10), $h_\beta$ is the recursive convolution of $2\beta + 2$ copies of $\mathbf{1}_{[-\frac{1}{2}, \frac{1}{2}]}$, and $\mathfrak{B}_\beta \triangleq \frac{1}{(\beta-1)!} \sum_{j=0}^{\lfloor \beta/2 \rfloor} (-1)^j \binom{\beta}{j} (\frac{\beta}{2} - j)^{\beta-1}$.

**Related work** For bounded $\mathbf{k}$, both i.i.d. samples (Tolstikhin et al., 2017, Prop. A.1) and thinned geometrically ergodic Markov chains (Dwivedi & Mackey, 2021, Prop. 1) deliver $n^{\frac{1}{2}}$ points with $\mathcal{O}(n^{-\frac{1}{4}})$ MMD with high probability. The *online Haar strategy* of Dwivedi et al. (2019) and low discrepancy *quasi-Monte Carlo* methods (see, e.g., Hickernell, 1998; Novak & Wozniakowski, 2010; Dick et al., 2013) provide improved $\mathcal{O}_d(n^{-\frac{1}{2}}\log^d n)$ MMD guarantees but are tailored specifically to the uniform distribution on $[0,1]^d$. Alternative coreset constructions for more general $\mathbb{P}$ include *kernel herding* (Chen et al., 2010), *discrepancy herding* (Harvey & Samadi, 2014), *super-sampling with a reservoir* (Paige et al., 2016), *support points convex-concave procedures* (Mak & Joseph, 2018), *greedy sign selection* (Karnin & Liberty, 2019, Sec. 3.1), *Stein point MCMC* (Chen et al., 2019), and *Stein thinning* (Riabiz et al., 2020a). While some admit better-than-i.i.d. MMD guarantees for finite-dimensional kernels on $\mathbb{R}^d$ (Chen et al., 2010; Harvey & Samadi, 2014), none

apart from KT are known to provide better-than-i.i.d. MMD or integration error for the infinite-dimensional kernels covered in this work. The lower bounds of Phillips & Tai (2020, Thm. 3.1) and Tolstikhin et al. (2017, Thm. 1) respectively establish that any procedure outputting $n^{\frac{1}{2}}$-sized core-sets and any procedure estimating $\mathbb{P}$ based only on $n$ i.i.d. sample points must incur $\Omega(n^{-\frac{1}{2}})$ MMD in the worst case. Our guarantees in Sec. 2 match these lower bounds up to logarithmic factors.

**Notation** We define the norm $\|\mathbf{k}\|_\infty = \sup_{x,y} |\mathbf{k}(x,y)|$ and the shorthand $[n] \triangleq \{1, \ldots, n\}$, $\mathbb{R}_+ \triangleq \{x \in \mathbb{R} : x \geq 0\}$, $\mathbb{N}_0 \triangleq \mathbb{N} \cup \{0\}$, $\mathcal{B}_\mathbf{k} \triangleq \{f \in \mathcal{H} : \|f\|_\mathbf{k} \leq 1\}$, and $\mathcal{B}_2(r) \triangleq \{y \in \mathbb{R}^d : \|y\|_2 \leq r\}$. We write $a \precsim b$ and $a \succsim b$ to mean $a = \mathcal{O}(b)$ and $a = \Omega(b)$, use $\precsim_d$ when masking constants dependent on $d$, and write $a = \mathcal{O}_p(b)$ to mean $a/b$ is bounded in probability. For any distribution $\mathbb{Q}$ and point sequences $\mathcal{S}, \mathcal{S}'$ with empirical distributions $\mathbb{Q}_n, \mathbb{Q}'_n$, we define $\mathrm{MMD}_\mathbf{k}(\mathbb{Q}, \mathcal{S}) \triangleq \mathrm{MMD}_\mathbf{k}(\mathbb{Q}, \mathbb{Q}_n)$ and $\mathrm{MMD}_\mathbf{k}(\mathcal{S}, \mathcal{S}') \triangleq \mathrm{MMD}_\mathbf{k}(\mathbb{Q}_n, \mathbb{Q}'_n)$.

## 2 GENERALIZED KERNEL THINNING

Our generalized kernel thinning algorithm (Alg. 1) for compressing an input point sequence $\mathcal{S}_{\mathrm{in}} = (x_i)_{i=1}^n$ proceeds in two steps: KT-SPLIT and KT-SWAP detailed in App. A. First, given a thinning parameter $m$ and an auxiliary kernel $\mathbf{k}_{\mathrm{split}}$, KT-SPLIT divides the input sequence into $2^m$ candidate coresets of size $n/2^m$ using non-uniform randomness. Next, given a target kernel $\mathbf{k}$, KT-SWAP selects a candidate coreset with smallest $\mathrm{MMD}_\mathbf{k}$ to $\mathcal{S}_{\mathrm{in}}$ and iteratively improves that coreset by exchanging coreset points for input points whenever the swap leads to reduced $\mathrm{MMD}_\mathbf{k}$. When $\mathbf{k}_{\mathrm{split}}$ is a square-root kernel $\mathbf{k}_{\mathrm{rt}}$ (2) of $\mathbf{k}$, generalized KT recovers the original ROOT KT algorithm of Dwivedi & Mackey. In this section, we establish performance guarantees for more general $\mathbf{k}_{\mathrm{split}}$ with special emphasis on the practical choice $\mathbf{k}_{\mathrm{split}} = \mathbf{k}$. Like ROOT KT, for any $m$, generalized KT has time complexity dominated by $\mathcal{O}(n^2)$ evaluations of $\mathbf{k}_{\mathrm{split}}$ and $\mathbf{k}$ and $\mathcal{O}(n \min(d, n))$ space complexity from storing either $\mathcal{S}_{\mathrm{in}}$ or the kernel matrices $(\mathbf{k}_{\mathrm{split}}(x_i, x_j))_{i,j=1}^n$ and $(\mathbf{k}(x_i, x_j))_{i,j=1}^n$.

---

**Algorithm 1:** Generalized Kernel Thinning – Return coreset of size $\lfloor n/2^m \rfloor$ with small $\mathrm{MMD}_\mathbf{k}$

**Input:** split kernel $\mathbf{k}_{\mathrm{split}}$, target kernel $\mathbf{k}$, point sequence $\mathcal{S}_{\mathrm{in}} = (x_i)_{i=1}^n$, thinning parameter $m \in \mathbb{N}$, probabilities $(\delta_i)_{i=1}^{\lfloor n/2 \rfloor}$

$(\mathcal{S}^{(m,\ell)})_{\ell=1}^{2^m} \leftarrow$ KT-SPLIT $(\mathbf{k}_{\mathrm{split}}, \mathcal{S}_{\mathrm{in}}, m, (\delta_i)_{i=1}^{\lfloor n/2 \rfloor})$    // Split $\mathcal{S}_{\mathrm{in}}$ into $2^m$ candidate coresets of size $\lfloor \frac{n}{2^m} \rfloor$

$\mathcal{S}_{\mathrm{KT}} \qquad\quad \leftarrow$ KT-SWAP $(\mathbf{k}, \mathcal{S}_{\mathrm{in}}, (\mathcal{S}^{(m,\ell)})_{\ell=1}^{2^m})$      // Select best coreset and iteratively refine

**return** coreset $\mathcal{S}_{\mathrm{KT}}$ of size $\lfloor n/2^m \rfloor$

---

### 2.1 SINGLE FUNCTION GUARANTEES FOR KT-SPLIT

We begin by analyzing the quality of the KT-SPLIT coresets. Our first main result, proved in App. B, bounds the KT-SPLIT integration error for any fixed function in the RKHS $\mathcal{H}_{\mathrm{split}}$ generated by $\mathbf{k}_{\mathrm{split}}$.

**Theorem 1 (Single function guarantees for KT-SPLIT)** *Consider* KT-SPLIT *(Alg. 1a) with oblivi-ous[3] $\mathcal{S}_{\mathrm{in}}$ and $(\delta_i)_{i=1}^{n/2}$ and $\delta^\star \triangleq \min_i \delta_i$. If $\frac{n}{2^m} \in \mathbb{N}$, then, for any fixed $f \in \mathcal{H}_{\mathrm{split}}$, index $\ell \in [2^m]$, and scalar $\delta' \in (0, 1)$, the output coreset $\mathcal{S}^{(m,\ell)}$ with $\mathbb{P}_{\mathrm{split}}^{(\ell)} \triangleq \frac{1}{n/2^m} \sum_{x \in \mathcal{S}^{(m,\ell)}} \boldsymbol{\delta}_x$ satisfies*

$$|\mathbb{P}_{\mathrm{in}} f - \mathbb{P}_{\mathrm{split}}^{(\ell)} f| \leq \|f\|_{\mathbf{k}_{\mathrm{split}}} \cdot \sigma_m \sqrt{2 \log(\tfrac{2}{\delta'})} \quad for \quad \sigma_m \triangleq \frac{2}{\sqrt{3}} \frac{2^m}{n} \sqrt{\|\mathbf{k}_{\mathrm{split}}\|_{\infty,\mathrm{in}} \cdot \log(\tfrac{6m}{2^m \delta^\star})}$$

*with probability at least $p_{\mathrm{sg}} \triangleq 1 - \delta' - \sum_{j=1}^m \frac{2^{j-1}}{m} \sum_{i=1}^{n/2^j} \delta_i$. Here, $\|\mathbf{k}_{\mathrm{split}}\|_{\infty,\mathrm{in}} \triangleq \max_{x \in \mathcal{S}_{\mathrm{in}}} \mathbf{k}_{\mathrm{split}}(x, x)$.*

**Remark 1 (Guarantees for known and oblivious stopping times)** *By Dwivedi & Mackey (2021, App. D), the success probability $p_{\mathrm{sg}}$ is at least $1 - \delta$ if we set $\delta' = \frac{\delta}{2}$ and $\delta_i = \frac{\delta}{n}$ for a stopping time $n$ known a priori or $\delta_i = \frac{m\delta}{2^{m+2}(i+1)\log^2(i+1)}$ for an arbitrary oblivious stopping time $n$.*

When compressing heavily from $n$ to $\sqrt{n}$ points, Thm. 1 and Rem. 1 guarantee $\mathcal{O}(n^{-\frac{1}{2}}\sqrt{\log n})$ inte-gration error with high probability for any fixed function $f \in \mathcal{H}_{\mathrm{split}}$. This represents a near-quadratic

---

[3]Throughout, *oblivious* indicates that a sequence is generated independently of any randomness in KT.

improvement over the $\Omega(n^{-\frac{1}{4}})$ integration error of $\sqrt{n}$ i.i.d. points. Moreover, this guarantee applies to **any kernel** defined on any space including unbounded kernels on unbounded domains (e.g., energy distance (Sejdinovic et al., 2013) and Stein kernels (Oates et al., 2017; Chwialkowski et al., 2016; Liu et al., 2016; Gorham & Mackey, 2017)); kernels with slowly decaying square roots (e.g., sinc kernels); and non-smooth kernels without square roots (e.g., Laplace, Matérn with $\gamma \in (\frac{d}{2}, d]$), and the compactly supported kernels of Wendland (2004) with $s < \frac{1}{2}(d+1)$). In contrast, the MMD guarantees of Dwivedi & Mackey covered only bounded, smooth $\mathbf{k}$ on $\mathbb{R}^d$ with bounded, Lipschitz, and rapidly-decaying square-roots. In addition, for $\|\mathbf{k}\|_\infty = 1$ on $\mathbb{R}^d$, the MMD bounds of Dwivedi & Mackey feature exponential dimension dependence of the form $c^d$ or $(\log n)^{d/2}$ while the Thm. 1 guarantee is **dimension-free** and hence practically relevant even when $d$ is large relative to $n$.

Thm. 1 also guarantees better-than-i.i.d. integration error for **any target distribution** with $|\mathbb{P}f - \mathbb{P}_{\text{in}}f| = o(n^{-\frac{1}{4}})$. In contrast, the MMD improvements of Dwivedi & Mackey (2021, cf. Tab. 2) applied only to $\mathbb{P}$ with at least $2d$ moments. Finally, when KT-SPLIT is applied with a square-root kernel $\mathbf{k}_{\text{split}} = \mathbf{k}_{\text{rt}}$, Thm. 1 still yields integration error bounds for $f \in \mathcal{H}$, as $\mathcal{H} \subseteq \mathcal{H}_{\text{split}}$. However, relative to target KT-SPLIT guarantees with $\mathbf{k}_{\text{split}} = \mathbf{k}$, the error bounds are inflated by a multiplicative factor of $\sqrt{\frac{\|\mathbf{k}_{\text{rt}}\|_{\infty,\text{in}}}{\|\mathbf{k}\|_{\infty,\text{in}}}} \frac{\|f\|_{\mathbf{k}_{\text{rt}}}}{\|f\|_{\mathbf{k}}}$. In App. H, we show that this inflation factor is at least 1 for each kernel explicitly analyzed in Dwivedi & Mackey (2021) and grows exponentially in dimension for Gaussian and Matérn kernels, unlike the dimension-free target KT-SPLIT bounds.

Finally, if we run KT-SPLIT with the perturbed kernel $\mathbf{k}'_{\text{split}}$ defined in Cor. 1, then we simultaneously obtain $\mathcal{O}(n^{-\frac{1}{2}}\sqrt{\log n})$ integration error for $f \in \mathcal{H}_{\text{split}}$, near-i.i.d. $\mathcal{O}(n^{-\frac{1}{4}}\sqrt{\log n})$ integration error for arbitrary bounded $f$ outside of $\mathcal{H}_{\text{split}}$, and intermediate, better-than-i.i.d. $o(n^{-\frac{1}{4}})$ integration error for smoother $f$ outside of $\mathcal{H}_{\text{split}}$ (by interpolation). We prove this guarantee in App. C.

**Corollary 1 (Guarantees for functions outside of $\mathcal{H}_{\text{split}}$)** *Consider extending each input point $x_i$ with the standard basis vector $e_i \in \mathbb{R}^n$ and running KT-SPLIT (Alg. 1a) on $\mathcal{S}'_{\text{in}} = (x_i, e_i)_{i=1}^n$ with $\mathbf{k}'_{\text{split}}((x,w),(y,v)) = \frac{\mathbf{k}_{\text{split}}(x,y)}{\|\mathbf{k}_{\text{split}}\|_\infty} + \langle w, v \rangle$ for $w, v \in \mathbb{R}^n$, $\in$. Under the notation and assumptions of Thm. 1, for any fixed index $\ell \in [2^m]$, scalar $\delta' \in (0,1)$, and $f$ defined on $\mathcal{S}_{\text{in}}$, we have, with probability at least $p_{\text{sg}}$,*

$$|\mathbb{P}_{\text{in}}f - \mathbb{P}^{(\ell)}_{\text{split}}f| \le \min(\sqrt{\tfrac{n}{2^m}}\|f\|_{\infty,\text{in}}, \sqrt{\|\mathbf{k}_{\text{split}}\|_\infty}\|f\|_{\mathbf{k}_{\text{split}}})\tfrac{2^m}{n}\sqrt{8\log(\tfrac{2}{\delta'})\cdot\log(\tfrac{8m}{2^m\delta^\star})}. \quad (3)$$

## 2.2 MMD GUARANTEE FOR TARGET KT

Our second main result bounds the $\text{MMD}_{\mathbf{k}}$ (1)—the worst-case integration error across the unit ball of $\mathcal{H}$—for generalized KT applied to the target kernel, i.e., $\mathbf{k}_{\text{split}} = \mathbf{k}$. The proof of this result in App. D is based on Thm. 1 and an appropriate covering number for the unit ball $\mathcal{B}_{\mathbf{k}}$ of the $\mathbf{k}$ RKHS.

**Definition 1 (k covering number)** *For a set $\mathcal{A}$ and scalar $\varepsilon > 0$, we define the $\mathbf{k}$ covering number $\mathcal{N}_{\mathbf{k}}(\mathcal{A}, \varepsilon)$ with $\mathcal{M}_{\mathbf{k}}(\mathcal{A}, \varepsilon) \triangleq \log \mathcal{N}_{\mathbf{k}}(\mathcal{A}, \varepsilon)$ as the minimum cardinality of a set $\mathcal{C} \subset \mathcal{B}_{\mathbf{k}}$ satisfying*

$$\mathcal{B}_{\mathbf{k}} \subseteq \bigcup_{h \in \mathcal{C}}\{g \in \mathcal{B}_{\mathbf{k}} : \sup_{x \in \mathcal{A}}|h(x) - g(x)| \le \varepsilon\}. \quad (4)$$

**Theorem 2 (MMD guarantee for TARGET KT)** *Consider generalized KT (Alg. 1) with $\mathbf{k}_{\text{split}} = \mathbf{k}$, oblivious $\mathcal{S}_{\text{in}}$ and $(\delta_i)_{i=1}^{\lfloor n/2 \rfloor}$, and $\delta^\star \triangleq \min_i \delta_i$. If $\frac{n}{2^m} \in \mathbb{N}$, then for any $\delta' \in (0,1)$, the output coreset $\mathcal{S}_{\text{KT}}$ is of size $\frac{n}{2^m}$ and satisfies*

$$\text{MMD}_{\mathbf{k}}(\mathcal{S}_{\text{in}}, \mathcal{S}_{\text{KT}}) \le \inf_{\varepsilon \in (0,1),\, \mathcal{S}_{\text{in}} \subset \mathcal{A}} 2\varepsilon + \tfrac{2^m}{n}\cdot\sqrt{\tfrac{8}{3}\|\mathbf{k}\|_{\infty,\text{in}}\log(\tfrac{6m}{2^m\delta^\star})\cdot\left[\log(\tfrac{4}{\delta'}) + \mathcal{M}_{\mathbf{k}}(\mathcal{A}, \varepsilon)\right]} \quad (5)$$

*with probability at least $p_{\text{sg}}$, where $\|\mathbf{k}\|_{\infty,\text{in}}$ and $p_{\text{sg}}$ were defined in Thm. 1.*

When compressing heavily from $n$ to $\sqrt{n}$ points, Thm. 2 and Rem. 1 with $\varepsilon = \sqrt{\frac{\|\mathbf{k}\|_{\infty,\text{in}}}{n}}$ and $\mathcal{A} = \mathcal{B}_2(\mathfrak{R}_{\text{in}})$ for $\mathfrak{R}_{\text{in}} \triangleq \max_{x \in \mathcal{S}_{\text{in}}}\|x\|_2$ guarantee

$$\text{MMD}_{\mathbf{k}}(\mathcal{S}_{\text{in}}, \mathcal{S}_{\text{KT}}) \precsim_\delta \sqrt{\tfrac{\|\mathbf{k}\|_{\infty,\text{in}}\log n}{n}\cdot\mathcal{M}_{\mathbf{k}}(\mathcal{B}_2(\mathfrak{R}_{\text{in}}), \sqrt{\tfrac{\|\mathbf{k}\|_{\infty,\text{in}}}{n}})} \quad (6)$$

with high probability. Thus we immediately obtain an MMD guarantee for any kernel $\mathbf{k}$ with a covering number bound. Furthermore, we readily obtain a comparable guarantee for $\mathbb{P}$ since $\mathrm{MMD}_{\mathbf{k}}(\mathbb{P}, \mathcal{S}_{\mathrm{KT}}) \le \mathrm{MMD}_{\mathbf{k}}(\mathbb{P}, \mathcal{S}_{\mathrm{in}}) + \mathrm{MMD}_{\mathbf{k}}(\mathcal{S}_{\mathrm{in}}, \mathcal{S}_{\mathrm{KT}})$. Any of a variety of existing algorithms can be used to generate an input point sequence $\mathcal{S}_{\mathrm{in}}$ with $\mathrm{MMD}_{\mathbf{k}}(\mathbb{P}, \mathcal{S}_{\mathrm{in}})$ no larger than the compression bound (6), including i.i.d. sampling (Tolstikhin et al., 2017, Thm. A.1), geometric MCMC (Dwivedi & Mackey, 2021, Prop. 1), kernel herding (Lacoste-Julien et al., 2015, Thm. G.1), Stein points (Chen et al., 2018, Thm. 2), Stein point MCMC (Chen et al., 2019, Thm. 1), greedy sign selection (Karnin & Liberty, 2019, Sec. 3.1), and Stein thinning (Riabiz et al., 2020a, Thm. 1).

### 2.3  Consequences of Thm. 2

Tab. 2 summarizes the MMD guarantees of Thm. 2 under common growth conditions on the log covering number $\mathcal{M}_{\mathbf{k}}$ and the input point radius $\mathfrak{R}_{\mathcal{S}_{\mathrm{in}}} \triangleq \max_{x \in \mathcal{S}_{\mathrm{in}}} \|x\|_2$. In Props. 2 and 3 of App. J, we show that analytic kernels, like Gaussian, inverse multiquadric (IMQ), and sinc, have **LogGrowth** $\mathcal{M}_{\mathbf{k}}$ (i.e., $\mathcal{M}_{\mathbf{k}}(\mathcal{B}_2(r), \varepsilon) \precsim_d r^d \log^{\omega}(\frac{1}{\varepsilon})$) while finitely differentiable kernels (like Matérn and B-spline) have **PolyGrowth** $\mathcal{M}_{\mathbf{k}}$ (i.e., $\mathcal{M}_{\mathbf{k}}(\mathcal{B}_2(r), \varepsilon) \precsim_d r^d \varepsilon^{-\omega}$).

Our conditions on $\mathfrak{R}_{\mathcal{S}_{\mathrm{in}}}$ arise from four forms of target distribution tail decay: (1) **Compact** ($\mathfrak{R}_{\mathcal{S}_{\mathrm{in}}} \precsim_d 1$), (2) **SubGauss** ($\mathfrak{R}_{\mathcal{S}_{\mathrm{in}}} \precsim_d \sqrt{\log n}$), (3) **SubExp** ($\mathfrak{R}_{\mathcal{S}_{\mathrm{in}}} \precsim_d \log n$), and (4) **HeavyTail** ($\mathfrak{R}_{\mathcal{S}_{\mathrm{in}}} \precsim_d n^{1/\rho}$). The first setting arises with a compactly supported $\mathbb{P}$ (e.g., on the unit cube $[0, 1]^d$), and the other three settings arise in expectation and with high probability when $\mathcal{S}_{\mathrm{in}}$ is generated i.i.d. from $\mathbb{P}$ with sub-Gaussian tails, sub-exponential tails, or $\rho$ moments respectively.

Substituting these conditions into (6) yields the eight entries of Tab. 2. We find that, for LogGrowth $\mathcal{M}_{\mathbf{k}}$, target KT MMD is within log factors of the $\Omega(n^{-1/2})$ lower bounds of Sec. 1 for light-tailed $\mathbb{P}$ and is $o(n^{-1/4})$ (i.e., better than i.i.d.) for any distribution with $\rho > 4d$ moments. Meanwhile, for PolyGrowth $\mathcal{M}_{\mathbf{k}}$, target KT MMD is $o(n^{-1/4})$ whenever $\omega < \frac{1}{2}$ for light-tailed $\mathbb{P}$ or whenever $\mathbb{P}$ has $\rho > 2d/(\frac{1}{2} - \omega)$ moments.

| | Compact $\mathbb{P}$
$\mathfrak{R}_{\mathrm{in}} \precsim_d 1$ | SubGauss $\mathbb{P}$
$\mathfrak{R}_{\mathrm{in}} \precsim_d \sqrt{\log n}$ | SubExp $\mathbb{P}$
$\mathfrak{R}_{\mathrm{in}} \precsim_d \log n$ | HeavyTail $\mathbb{P}$
$\mathfrak{R}_{\mathrm{in}} \precsim_d n^{1/\rho}$ |
|---|---|---|---|---|
| LogGrowth $\mathcal{M}_{\mathbf{k}}$
$\mathcal{M}_{\mathbf{k}}(\mathcal{B}_2(r), \varepsilon) \precsim_d r^d \log^{\omega}(\frac{1}{\varepsilon})$ | $\sqrt{\frac{(\log n)^{\omega+1}}{n}}$ | $\sqrt{\frac{(\log n)^{d+\omega+1}}{n}}$ | $\sqrt{\frac{(\log n)^{2d+\omega+1}}{n}}$ | $\sqrt{\frac{(\log n)^{\omega+1}}{n^{1-2d/\rho}}}$ |
| PolyGrowth $\mathcal{M}_{\mathbf{k}}$
$\mathcal{M}_{\mathbf{k}}(\mathcal{B}_2(r), \varepsilon) \precsim_d r^d \varepsilon^{-\omega}$ | $\sqrt{\frac{\log n}{n^{1-\omega}}}$ | $\sqrt{\frac{(\log n)^{d+1}}{n^{1-\omega}}}$ | $\sqrt{\frac{(\log n)^{2d+1}}{n^{1-\omega}}}$ | $\sqrt{\frac{\log n}{n^{1-\omega-2d/\rho}}}$ |

**Table 2: MMD guarantees for target KT under $\mathcal{M}_{\mathbf{k}}$ (4) growth and $\mathbb{P}$ tail decay.** We report the $\mathrm{MMD}_{\mathbf{k}}(\mathcal{S}_{\mathrm{in}}, \mathcal{S}_{\mathrm{KT}})$ bound (6) for target KT with $n$ input points and $\sqrt{n}$ output points, up to constants depending on $d$ and $\|\mathbf{k}\|_{\infty, \mathrm{in}}$. Here $\mathfrak{R}_{\mathrm{in}} \triangleq \max_{x \in \mathcal{S}_{\mathrm{in}}} \|x\|_2$.

Next, for each of the popular convergence-determining kernels of Tab. 1, we compare the root KT MMD guarantees of Dwivedi & Mackey (2021) with the target KT guarantees of Thm. 2 combined with covering number bounds derived in Apps. J and K. We see in Tab. 3 that Thm. 2 provides better-than-i.i.d. and better-than-root KT guarantees for kernels with slowly decaying or non-existent square-roots (e.g., IMQ with $\nu < \frac{d}{2}$, sinc, and B-spline) and nearly matches known root KT guarantees for analytic kernels like Gauss and IMQ with $\nu \ge \frac{d}{2}$, even though target KT makes no explicit use of a square-root kernel. See App. K for the proofs related to Tab. 3.

## 3  Kernel Thinning+

We next introduce and analyze two new generalized KT variants: (i) power KT which leverages a power kernel $\mathbf{k}_\alpha$ that interpolates between $\mathbf{k}$ and $\mathbf{k}_{\mathrm{rt}}$ to improve upon the MMD guarantees of target KT even when $\mathbf{k}_{\mathrm{rt}}$ is not available and (ii) KT+ which uses a sum of $\mathbf{k}$ and $\mathbf{k}_\alpha$ to retain both the improved MMD guarantee of $\mathbf{k}_\alpha$ and the superior single function guarantees of $\mathbf{k}$.

**Power kernel thinning**  First, we generalize the square-root kernel (2) definition for shift-invariant $\mathbf{k}$ using the order 0 generalized Fourier transform (GFT, Wendland, 2004, Def. 8.9) $\widehat{f}$ of $f : \mathbb{R}^d \to \mathbb{R}$.

| Kernel k | TARGET KT | ROOT KT | KT+ |
|---|---|---|---|
| GAUSS($\sigma$) | $\frac{(\log n)^{\frac{3d}{4}+1}}{\sqrt{n \cdot c_n^d}}$ | $\frac{(\log n)^{\frac{d}{4}+\frac{1}{2}}\sqrt{c_n}}{\sqrt{n}}$ | $\frac{(\log n)^{\frac{d}{4}+\frac{1}{2}}\sqrt{c_n}}{\sqrt{n}}$ |
| LAPLACE($\sigma$) | $n^{-\frac{1}{4}}$ | N/A | $\left(\frac{c_n(\log n)^{1+2d(1-\alpha)}}{n}\right)^{\frac{1}{4\alpha}}$ |
| MATÉRN($\nu,\gamma$): $\nu \in (\frac{d}{2}, d]$ | $n^{-\frac{1}{4}}$ | N/A | $\left(\frac{c_n(\log n)^{1+2d(1-\alpha)}}{n}\right)^{\frac{1}{4\alpha}}$ |
| MATÉRN($\nu,\gamma$): $\nu > d$ | $\min(n^{-\frac{1}{4}}, \frac{(\log n)^{\frac{d+1}{2}}}{n^{(\nu-d)/(2\nu-d)}})$ | $\frac{(\log n)^{\frac{d+1}{2}}\sqrt{c_n}}{\sqrt{n}}$ | $\frac{(\log n)^{\frac{d+1}{2}}\sqrt{c_n}}{\sqrt{n}}$ |
| IMQ($\nu,\gamma$): $\nu < \frac{d}{2}$ | $\frac{(\log n)^{d+1}}{\sqrt{n}}$ | $n^{-\frac{1}{4}}$ | $\frac{(\log n)^{d+1}}{\sqrt{n}}$ |
| IMQ($\nu,\gamma$): $\nu \geq \frac{d}{2}$ | $\frac{(\log n)^{d+1}}{\sqrt{n}}$ | $\frac{(\log n)^{\frac{d+1}{2}}\sqrt{c_n}}{\sqrt{n}}$ | $\frac{(\log n)^{\frac{d+1}{2}}\sqrt{c_n}}{\sqrt{n}}$ |
| SINC($\theta$) | $\frac{(\log n)^2}{\sqrt{n}}$ | $n^{-\frac{1}{4}}$ | $\frac{(\log n)^2}{\sqrt{n}}$ |
| B-SPLINE($2\beta+1,\gamma$): $\beta \in 2\mathbb{N}$ | $\sqrt{\frac{\log n}{n^{2\beta/(2\beta+1)}}}$ | N/A | $\sqrt{\frac{\log n}{n}}$ |
| B-SPLINE($2\beta+1,\gamma$): $\beta \in 2\mathbb{N}_0 + 1$ | $\sqrt{\frac{\log n}{n^{2\beta/(2\beta+1)}}}$ | $\sqrt{\frac{\log n}{n}}$ | $\sqrt{\frac{\log n}{n}}$ |

**Table 3:** $\mathrm{MMD}_\mathbf{k}(\mathcal{S}_\mathrm{in}, \mathcal{S}_\mathrm{KT})$ **guarantees for commonly used kernels.** For $n$ input and $\sqrt{n}$ output points, we report the MMD bounds of Thm. 2 for TARGET KT, of Dwivedi & Mackey (2021, Thm. 1) for ROOT KT, and of Thm. 4 for KT+ (with $\alpha = \frac{1}{2}$ wherever feasible). We assume a SUBGAUSS $\mathbb{P}$ for the GAUSS kernel, a COMPACT $\mathbb{P}$ for the B-SPLINE kernel, and a SUBEXP $\mathbb{P}$ for all other $\mathbf{k}$ (see Tab. 2 for a definition of each $\mathbb{P}$ class). Here, $c_n \triangleq \log\log n$, $\delta_i = \frac{\delta}{n}$, $\delta' = \frac{\delta}{2}$, and error is reported up to constants depending on $(\mathbf{k}, d, \delta, \alpha)$. The KT+ guarantee for LAPLACE applies with $\alpha > \frac{d}{d+1}$ and for MATÉRN with $\alpha > \frac{d}{2\nu}$. The TARGET KT guarantee for MATÉRN with $\nu > 3d/2$ assumes $\nu - d/2 \in \mathbb{N}$ to simplify the presentation (see (53) for the general case). The best rate is highlighted in **blue**.

**Definition 2 ($\alpha$-power kernel)** *Define* $\mathbf{k}_1 \triangleq \mathbf{k}$. *We say a kernel* $\mathbf{k}_{\frac{1}{2}}$ *is a* $\frac{1}{2}$*-power kernel for* $\mathbf{k}$ *if* $\mathbf{k}(x,y) = (2\pi)^{-d/2}\int_{\mathbb{R}^d}\mathbf{k}_{\frac{1}{2}}(x,z)\mathbf{k}_{\frac{1}{2}}(z,y)dz$. *For* $\alpha \in (\frac{1}{2}, 1)$, *a kernel* $\mathbf{k}_\alpha(x,y) = \kappa_\alpha(x-y)$ *on* $\mathbb{R}^d$ *is an* $\alpha$*-power kernel for* $\mathbf{k}(x,y) = \kappa(x-y)$ *if* $\widehat{\kappa_\alpha} = \widehat{\kappa}^\alpha$.

By design, $\mathbf{k}_{\frac{1}{2}}$ matches $\mathbf{k}_\mathrm{rt}$ (2) up to an immaterial constant rescaling. Given a power kernel $\mathbf{k}_\alpha$ we define POWER KT as generalized KT with $\mathbf{k}_\mathrm{split} = \mathbf{k}_\alpha$. Our next result (with proof in App. E) provides an MMD guarantee for POWER KT.

**Theorem 3 (MMD guarantee for POWER KT)** *Consider generalized KT (Alg. 1) with* $\mathbf{k}_\mathrm{split} = \mathbf{k}_\alpha$ *for some* $\alpha \in [\frac{1}{2}, 1]$, *oblivious sequences* $\mathcal{S}_\mathrm{in}$ *and* $(\delta_i)_{i=1}^{\lfloor n/2\rfloor}$, *and* $\delta^\star \triangleq \min_i \delta_i$. *If* $\frac{n}{2^m} \in \mathbb{N}$, *then for any* $\delta' \in (0,1)$, *the output coreset* $\mathcal{S}_\mathrm{KT}$ *is of size* $\frac{n}{2^m}$ *and satisfies*

$$\mathrm{MMD}_\mathbf{k}(\mathcal{S}_\mathrm{in}, \mathcal{S}_\mathrm{KT}) \leq \left(\frac{2^m}{n}\|\mathbf{k}_\alpha\|_\infty\right)^{\frac{1}{2\alpha}}(2\cdot\widetilde{\mathfrak{M}}_\alpha)^{1-\frac{1}{2\alpha}}\left(2+\sqrt{\frac{(4\pi)^{d/2}}{\Gamma(\frac{d}{2}+1)}\cdot\mathfrak{R}_\mathrm{max}^{\frac{d}{2}}\cdot\widetilde{\mathfrak{M}}_\alpha}\right)^{\frac{1}{\alpha}-1}, \quad (7)$$

*with probability at least* $p_\mathrm{sg}$ *(defined in Thm. 1). The parameters* $\widetilde{\mathfrak{M}}_\alpha$ *and* $\mathfrak{R}_\mathrm{max}$ *are defined in App. E and satisfy* $\widetilde{\mathfrak{M}}_\alpha = \mathcal{O}_d(\sqrt{\log n})$ *and* $\mathfrak{R}_\mathrm{max} = \mathcal{O}_d(1)$ *for compactly supported* $\mathbb{P}$ *and* $\mathbf{k}_\alpha$ *and* $\widetilde{\mathfrak{M}}_\alpha = \mathcal{O}_d(\sqrt{\log n \log\log n})$ *and* $\mathfrak{R}_\mathrm{max} = \mathcal{O}_d(\log n)$ *for subexponential* $\mathbb{P}$ *and* $\mathbf{k}_\alpha$, *when* $\delta^\star = \frac{\delta'}{n}$.

Thm. 3 reproduces the ROOT KT guarantee of Dwivedi & Mackey (2021, Thm. 1) when $\alpha = \frac{1}{2}$ and more generally accommodates any power kernel via an MMD interpolation result (Prop. 1) that may be of independent interest. This generalization is especially valuable for less-smooth kernels like LAPLACE and MATÉRN($\nu,\gamma$) with $\nu \in (\frac{d}{2}, d]$ that have no square-root kernel. Our TARGET KT MMD guarantees are no better than i.i.d. for these kernels, but, as shown in App. K, these kernels have MATÉRN kernels as $\alpha$-power kernels, which yield $o(n^{-\frac{1}{4}})$ MMD in conjunction with Thm. 3.

**Kernel thinning+** Our final KT variant, *kernel thinning+*, runs KT-SPLIT with a scaled sum of the target and power kernels, $\mathbf{k}^\dagger \triangleq \mathbf{k}/\|\mathbf{k}\|_\infty + \mathbf{k}_\alpha/\|\mathbf{k}_\alpha\|_\infty$.[4] Remarkably, this choice simultaneously provides the improved MMD guarantees of Thm. 3 and the dimension-free single function guarantees of Thm. 1 (see App. F for the proof).

---

[4] When $\mathcal{S}_\mathrm{in}$ is known in advance, one can alternatively choose $\mathbf{k}^\dagger \triangleq \mathbf{k}/\|\mathbf{k}\|_{\infty,\mathrm{in}} + \mathbf{k}_\alpha/\|\mathbf{k}_\alpha\|_{\infty,\mathrm{in}}$.

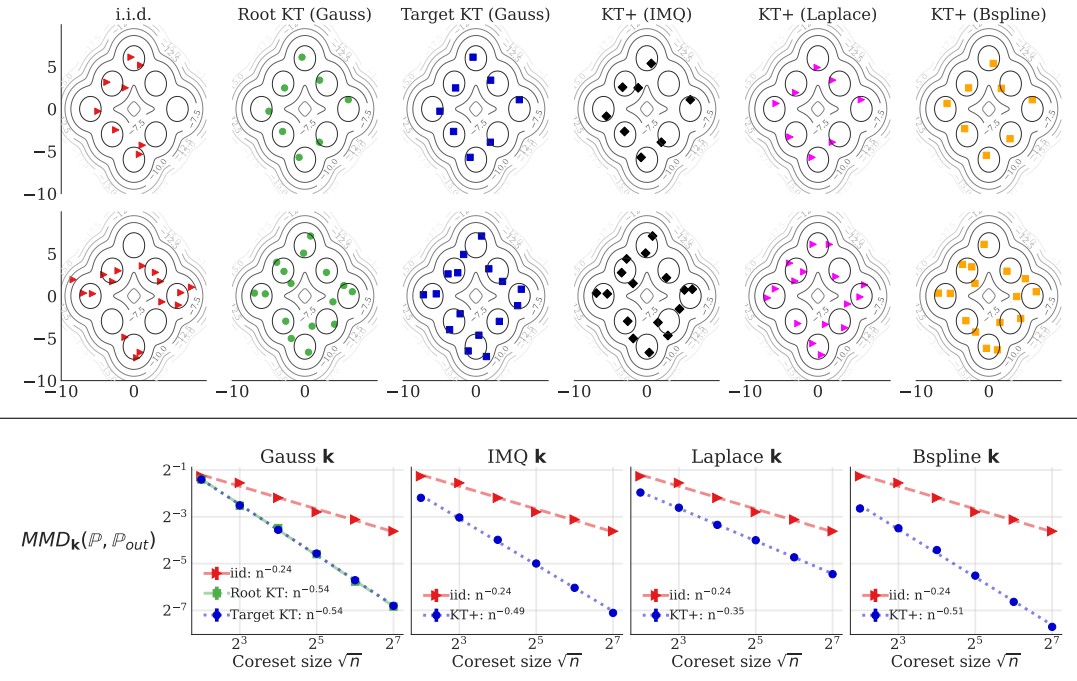

**Figure 1: Generalized kernel thinning (KT) vs i.i.d. sampling** for an 8-component mixture of Gaussians target $\mathbb{P}$. For kernels $\mathbf{k}$ without fast-decaying square-roots, KT+ offers visible and quantifiable improvements over i.i.d. sampling. For Gaussian $\mathbf{k}$, TARGET KT closely mimics ROOT KT.

**Theorem 4 (Single function & MMD guarantees for KT+)** *Consider generalized KT (Alg. 1) with $\mathbf{k}_{\mathrm{split}} = \mathbf{k}^\dagger$, oblivious $\mathcal{S}_{\mathrm{in}}$ and $(\delta_i)_{i=1}^{\lfloor n/2 \rfloor}$, $\delta^\star \triangleq \min_i \delta_i$, and $\frac{n}{2^m} \in \mathbb{N}$. For any fixed function $f \in \mathcal{H}$, index $\ell \in [2^m]$, and scalar $\delta' \in (0,1)$, the KT-SPLIT coreset $\mathcal{S}^{(m,\ell)}$ satisfies*

$$|\mathbb{P}_{\mathrm{in}} f - \mathbb{P}_{\mathrm{split}}^{(\ell)} f| \leq \frac{2^m}{n} \cdot \sqrt{\frac{16}{3} \log(\frac{6m}{2^m \delta^\star}) \log(\frac{2}{\delta'})} \|f\|_{\mathbf{k}} \sqrt{\|\mathbf{k}\|_\infty}, \tag{8}$$

*with probability at least $p_{\mathrm{sg}}$ (for $p_{\mathrm{sg}}$ and $\mathbb{P}_{\mathrm{split}}^{(\ell)}$ defined in Thm. 1). Moreover,*

$$\mathrm{MMD}_{\mathbf{k}}(\mathcal{S}_{\mathrm{in}}, \mathcal{S}_{\mathrm{KT}}) \leq \min\left[\sqrt{2} \cdot \overline{\mathbf{M}}_{\mathrm{targetKT}}(\mathbf{k}), \quad 2^{\frac{1}{2\alpha}} \cdot \overline{\mathbf{M}}_{\mathrm{powerKT}}(\mathbf{k}_\alpha)\right] \tag{9}$$

*with probability at least $p_{\mathrm{sg}}$, where $\overline{\mathbf{M}}_{\mathrm{targetKT}}(\mathbf{k})$ denotes the right hand side of (5) with $\|\mathbf{k}\|_{\infty,\mathrm{in}}$ replaced by $\|\mathbf{k}\|_\infty$, and $\overline{\mathbf{M}}_{\mathrm{powerKT}}(\mathbf{k}_\alpha)$ denotes the right hand side of (7).*

As shown in Tab. 3, KT+ provides better-than-i.i.d. MMD guarantees for every kernel in Tab. 1—even the Laplace, non-smooth Matérn, and odd B-spline kernels neglected by prior analyses—while matching or improving upon the guarantees of TARGET KT and ROOT KT in each case.

## 4 EXPERIMENTS

Dwivedi & Mackey (2021) illustrated the MMD benefits of ROOT KT over i.i.d. sampling and standard MCMC thinning with a series of vignettes focused on the Gaussian kernel. We revisit those vignettes with the broader range of kernels covered by generalized KT and demonstrate significant improvements in both MMD and single-function integration error. We focus on coresets of size $\sqrt{n}$ produced from $n$ inputs with $\delta_i = \frac{1}{2n}$, let $\mathbb{P}_{\mathrm{out}}$ denote the empirical distribution of each output coreset, and report mean error ($\pm 1$ standard error) over 10 independent replicates of each experiment.

**Target distributions and kernel bandwidths** We consider three classes of target distributions on $\mathbb{R}^d$: (i) mixture of Gaussians $\mathbb{P} = \frac{1}{M} \sum_{j=1}^M \mathcal{N}(\mu_j, \mathbf{I}_2)$ with $M$ component means $\mu_j \in \mathbb{R}^2$ defined in App. I, (ii) Gaussian $\mathbb{P} = \mathcal{N}(0, \mathbf{I}_d)$, and (iii) the posteriors of four distinct coupled ordinary

differential equation models: the *Goodwin (1965) model* of oscillatory enzymatic control ($d = 4$), the *Lotka (1925) model* of oscillatory predator-prey evolution ($d = 4$), the *Hinch et al. (2004) model* of calcium signalling in cardiac cells ($d = 38$), and a tempered Hinch posterior. For settings (i) and (ii), we use an i.i.d. input sequence $\mathcal{S}_{\text{in}}$ from $\mathbb{P}$ and kernel bandwidths $\sigma = 1/\gamma = \sqrt{2d}$. For setting (iii), we use MCMC input sequences $\mathcal{S}_{\text{in}}$ from 12 posterior inference experiments of Riabiz et al. (2020a) and set the bandwidths $\sigma = 1/\gamma$ as specified by Dwivedi & Mackey (2021, Sec. K.2).

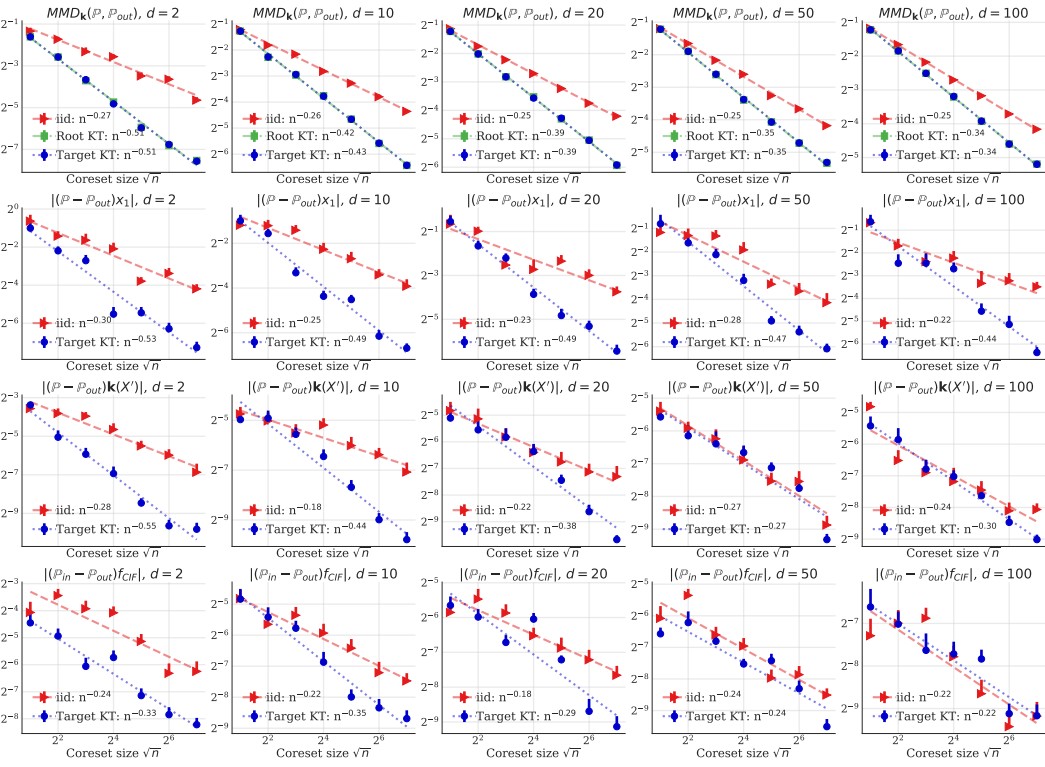

**Figure 2: MMD and single-function integration error for Gaussian k and standard Gaussian $\mathbb{P}$ in $\mathbb{R}^d$.** Without using a square-root kernel, TARGET KT matches the MMD performance of ROOT KT and improves upon i.i.d. MMD and single-function integration error, even in $d = 100$ dimensions.

**Function testbed** To evaluate the ability of generalized KT to improve integration both inside and outside of $\mathcal{H}$, we evaluate integration error for (a) a random element of the target kernel RKHS ($f(x) = \mathbf{k}(X', x)$ described in App. I), (b) moments ($f(x) = x_1$ and $f(x) = x_1^2$), and (c) a standard numerical integration benchmark test function from the *continuous integrand family* (CIF, Genz, 1984), $f_{\text{CIF}}(x) = \exp(-\frac{1}{d}\sum_{j=1}^{d}|x_j - u_j|)$ for $u_j$ drawn i.i.d. and uniformly from $[0, 1]$.

**Generalized KT coresets** For an 8-component mixture of Gaussians target $\mathbb{P}$, the top row of Fig. 1 highlights the visual differences between i.i.d. coresets and coresets generated using generalized KT. We consider ROOT KT with GAUSS **k**, TARGET KT with GAUSS **k**, and KT+ ($\alpha = 0.7$) with LAPLACE **k**, KT+ ($\alpha = \frac{1}{2}$) with IMQ **k** ($\nu = 0.5$), and KT+($\alpha = \frac{2}{3}$) with B-SPLINE(5) **k**, and note that the B-SPLINE(5) and LAPLACE **k** do not admit square-root kernels. In each case, even for small $n$, generalized KT provides a more even distribution of points across components with fewer within-component gaps and clumps. Moreover, as suggested by our theory, TARGET KT and ROOT KT coresets for GAUSS **k** have similar quality despite TARGET KT making no explicit use of a square-root kernel. The MMD error plots in the bottom row of Fig. 1 provide a similar conclusion quantitatively, where we observe that for both variants of KT, the MMD error decays as $n^{-\frac{1}{2}}$, a significant improvement over the $n^{-\frac{1}{4}}$ rate of i.i.d. sampling. We also observe that the empirical MMD decay rates are in close agreement with the rates guaranteed by our theory in Tab. 3 ($n^{-\frac{1}{2}}$ for GAUSS, B-SPLINE, and IMQ and $n^{-\frac{1}{4\alpha}} = n^{-0.36}$ for LAPLACE). We provide additional visualizations and results in Figs. 4 and 5 of App. I, including MMD errors for $M = 4$ and $M = 6$ component mixture targets. The conclusions remain consistent with those drawn from Fig. 1.

**TARGET KT vs. i.i.d. sampling** For Gaussian $\mathbb{P}$ and Gaussian $\mathbf{k}$, Fig. 2 quantifies the improvements in distributional approximation obtained when using TARGET KT in place of a more typical i.i.d. summary. Remarkably, TARGET KT significantly improves the rate of decay and order of magnitude of mean $\mathrm{MMD}_{\mathbf{k}}(\mathbb{P}, \mathbb{P}_{\mathrm{out}})$, even in $d = 100$ dimensions with as few as 4 output points. Moreover, in line with our theory, TARGET KT MMD closely tracks that of ROOT KT without using $\mathbf{k}_{\mathrm{rt}}$. Finally, TARGET KT delivers improved single-function integration error, both of functions in the RKHS (like $\mathbf{k}(X', \cdot)$) and those outside (like the first moment and CIF benchmark function), even with large $d$ and relatively small sample sizes.

**KT+ vs. standard MCMC thinning** For the MCMC targets, we measure error with respect to the input distribution $\mathbb{P}_{\mathrm{in}}$ (consistent with our guarantees), as exact integration under each posterior $\mathbb{P}$ is intractable. We employ KT+ ($\alpha = 0.81$) with LAPLACE $\mathbf{k}$ for Goodwin and Lotka-Volterra and KT+ ($\alpha = 0.5$) with IMQ $\mathbf{k}$ ($\nu = 0.5$) for Hinch. Notably, neither kernel has a square-root with fast-decaying tails. In Fig. 3, we evaluate thinning results from one chain targeting each of the Goodwin, Lotka-Volterra, and Hinch posteriors and observe that KT+ uniformly improves upon the MMD error of standard thinning (ST), even when ST exhibits better-than-i.i.d. accuracy. Furthermore, KT+ provides significantly smaller integration error for functions inside of the RKHS (like $\mathbf{k}(X', \cdot)$) and outside of the RKHS (like the first and second moments and the benchmark CIF function) in nearly every setting. See Fig. 6 of App. I for plots of the other 9 MCMC settings.

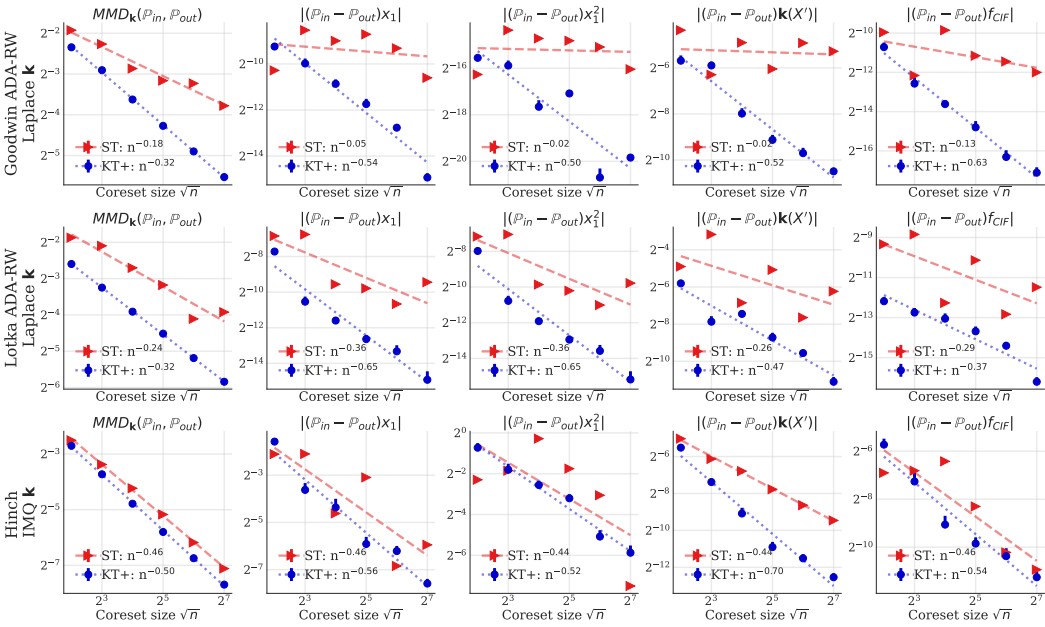

**Figure 3: Kernel thinning+ (KT+) vs. standard MCMC thinning (ST).** For kernels without fast-decaying square-roots, KT+ improves MMD and integration error decay rates in each posterior inference task.

## 5 DISCUSSION AND CONCLUSIONS

In this work, we introduced three new generalizations of the ROOT KT algorithm of Dwivedi & Mackey (2021) with broader applicability and strengthened guarantees for generating compact representations of a probability distribution. TARGET KT-SPLIT provides $\sqrt{n}$-point summaries with $\mathcal{O}(\sqrt{\log n/n})$ integration error guarantees for any kernel, any target distribution, and any function in the RKHS; POWER KT yields improved better-than-i.i.d. MMD guarantees even when a square-root kernel is unavailable; and KT+ simultaneously inherits the guarantees of TARGET KT and POWER KT. While we have focused on unweighted coreset quality we highlight that the same MMD guarantees extend to any improved reweighting of the coreset points. For example, for downstream tasks that support weights, one can optimally reweight $\mathbb{P}_{\mathrm{out}}$ to approximate $\mathbb{P}_{\mathrm{in}}$ in $\mathcal{O}(n^{\frac{3}{2}})$ time by directly minimizing $\mathrm{MMD}_{\mathbf{k}}$. Finally, one can combine generalized KT with the COMPRESS++ meta-algorithm of Shetty et al. (2022) to obtain coresets of comparable quality in near-linear time.

REPRODUCIBILITY STATEMENT

See App. I for supplementary experimental details and results and the `goodpoints` Python package

$$\texttt{https://github.com/microsoft/goodpoints}$$

for Python code reproducing all experiments.

ACKNOWLEDGMENTS

We thank Lucas Janson and Boaz Barak for their valuable feedback on this work. RD acknowledges the support by the National Science Foundation under Grant No. DMS-2023528 for the Foundations of Data Science Institute (FODSI).

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

## APPENDIX

## A  DETAILS OF KT-SPLIT AND KT-SWAP

---

**Algorithm 1a:** KT-SPLIT $-$ Divide points into candidate coresets of size $\lfloor n/2^m \rfloor$

---

**Input:** kernel $\mathbf{k}_{\mathrm{split}}$, point sequence $\mathcal{S}_{\mathrm{in}} = (x_i)_{i=1}^n$, thinning parameter $m \in \mathbb{N}$, probabilities $(\delta_i)_{i=1}^{\lfloor \frac{n}{2} \rfloor}$

$\mathcal{S}^{(j,\ell)} \leftarrow \{\}$ for $0 \le j \le m$ and $1 \le \ell \le 2^j$    // Empty coresets: $\mathcal{S}^{(j,\ell)}$ has size $\lfloor \frac{i}{2^j} \rfloor$ after round $i$

$\sigma_{j,\ell} \leftarrow 0$ for $1 \le j \le m$ and $1 \le \ell \le 2^{j-1}$    // Swapping parameters

**for** $i = 1, \ldots, \lfloor n/2 \rfloor$ **do**

$\quad$ $\mathcal{S}^{(0,1)}$.append$(x_i)$; $\mathcal{S}^{(0,1)}$.append$(x_{2i})$

$\quad$ // Every $2^j$ rounds, add one point from parent coreset $\mathcal{S}^{(j-1,\ell)}$ to each child coreset $\mathcal{S}^{(j,2\ell-1)}, \mathcal{S}^{(j,2\ell)}$

$\quad$ **for** $(j = 1;\ j \le m$ **and** $i/2^{j-1} \in \mathbb{N};\ j = j+1)$ **do**

$\quad\quad$ **for** $\ell = 1, \ldots, 2^{j-1}$ **do**

$\quad\quad\quad$ $(\mathcal{S}, \mathcal{S}') \leftarrow (\mathcal{S}^{(j-1,\ell)}, \mathcal{S}^{(j,2\ell-1)});\quad (x, \tilde{x}) \leftarrow$ get_last_two_points$(\mathcal{S})$

$\quad\quad\quad$ // Compute swapping threshold $\mathfrak{a}$

$\quad\quad\quad$ $\mathfrak{a}, \sigma_{j,\ell} \leftarrow$ get_swap_params$(\sigma_{j,\ell}, \mathfrak{b}, \delta_{|\mathcal{S}|/2} \frac{2^{j-1}}{m})$ for $\mathfrak{b}^2 = \mathbf{k}_{\mathrm{split}}(x,x) + \mathbf{k}_{\mathrm{split}}(\tilde{x}, \tilde{x}) - 2\mathbf{k}_{\mathrm{split}}(x, \tilde{x})$

$\quad\quad\quad$ // Assign one point to each child after probabilistic swapping

$\quad\quad\quad$ $\alpha \leftarrow \mathbf{k}_{\mathrm{split}}(\tilde{x}, \tilde{x}) - \mathbf{k}_{\mathrm{split}}(x,x) + \Sigma_{y \in \mathcal{S}}(\mathbf{k}_{\mathrm{split}}(y,x) - \mathbf{k}_{\mathrm{split}}(y, \tilde{x})) - 2\Sigma_{z \in \mathcal{S}'}(\mathbf{k}_{\mathrm{split}}(z,x) - \mathbf{k}_{\mathrm{split}}(z, \tilde{x}))$

$\quad\quad\quad$ $(x, \tilde{x}) \leftarrow (\tilde{x}, x)$ *with probability* $\min(1, \frac{1}{2}(1 - \frac{\alpha}{\mathfrak{a}})_+)$

$\quad\quad\quad$ $\mathcal{S}^{(j,2\ell-1)}$.append$(x);\quad \mathcal{S}^{(j,2\ell)}$.append$(\tilde{x})$

$\quad\quad$ **end**

$\quad$ **end**

**end**

**return** $(\mathcal{S}^{(m,\ell)})_{\ell=1}^{2^m}$, candidate coresets of size $\lfloor n/2^m \rfloor$

---

**function** get_swap_params$(\sigma, \mathfrak{b}, \delta)$:

$\quad$ $\mathfrak{a} \leftarrow \max(\mathfrak{b}\sigma\sqrt{2\log(2/\delta)}, \mathfrak{b}^2)$

$\quad$ $\sigma \leftarrow \sigma + \mathfrak{b}^2(1 + (\mathfrak{b}^2 - 2\mathfrak{a})\sigma^2/\mathfrak{a}^2)_+$

**return** $(\mathfrak{a}, \sigma)$

---

---

**Algorithm 1b:** KT-SWAP − Identify and refine the best candidate coreset

---

**Input:** kernel $\mathbf{k}$, point sequence $\mathcal{S}_{\text{in}} = (x_i)_{i=1}^n$, candidate coresets $(\mathcal{S}^{(m,\ell)})_{\ell=1}^{2^m}$

$\mathcal{S}^{(m,0)} \leftarrow \texttt{baseline\_thinning}(\mathcal{S}_{\text{in}}, \texttt{size} = \lfloor n/2^m \rfloor)$   // Compare to baseline (e.g., standard thinning)

$\mathcal{S}_{\text{KT}} \leftarrow \mathcal{S}^{(m,\ell^\star)}$ for $\ell^\star \leftarrow \operatorname{argmin}_{\ell \in \{0,1,\ldots,2^m\}} \operatorname{MMD}_{\mathbf{k}}(\mathcal{S}_{\text{in}}, \mathcal{S}^{(m,\ell)})$ // Select best candidate coreset

// Swap out each point in $\mathcal{S}_{\text{KT}}$ for best alternative in $\mathcal{S}_{\text{in}}$

**for** $i = 1, \ldots, \lfloor n/2^m \rfloor$ **do**

$\quad \mathcal{S}_{\text{KT}}[i] \leftarrow \operatorname{argmin}_{z \in \mathcal{S}_{\text{in}}} \operatorname{MMD}_{\mathbf{k}}(\mathcal{S}_{\text{in}}, \mathcal{S}_{\text{KT}} \text{ with } \mathcal{S}_{\text{KT}}[i] = z)$

**end**

**return** $\mathcal{S}_{\text{KT}}$, refined coreset of size $\lfloor n/2^m \rfloor$

---

## B    Proof of Thm. 1: Single function guarantees for KT-SPLIT

The proof is identical for each index $\ell$, so, without loss of generality, we prove the result for the case $\ell = 1$. Define

$$\widetilde{\mathcal{W}}_m \triangleq \mathcal{W}_{1,m} = \mathbb{P}_{\text{in}}\mathbf{k}_{\text{split}} - \mathbb{P}_{\text{out}}^{(1)}\mathbf{k}_{\text{split}} = \frac{1}{n}\sum_{x \in \mathcal{S}_{\text{in}}} \mathbf{k}_{\text{split}}(x, \cdot) - \frac{1}{n/2^m}\sum_{x \in \mathcal{S}^{(m,1)}} \mathbf{k}_{\text{split}}(x, \cdot).$$

Next, we use the results about an intermediate algorithm, kernel halving (Dwivedi & Mackey, 2021, Alg. 3) that was introduced for the analysis of kernel thinning. Using the arguments from Dwivedi & Mackey (2021, Sec. 5.2), we conclude that KT-SPLIT with $\mathbf{k}_{\text{split}}$ and thinning parameter $m$, is equivalent to repeated kernel halving with kernel $\mathbf{k}_{\text{split}}$ for $m$ rounds (with no Failure in any rounds of kernel halving). On this event of equivalence, denoted by $\mathcal{E}_{\text{equi}}$, Dwivedi & Mackey (2021, Eqns. (50, 51)) imply that the function $\widetilde{\mathcal{W}}_m \in \mathcal{H}_{\text{split}}$ is equal in distribution to another random function $\mathcal{W}_m$, where $\mathcal{W}_m$ is unconditionally sub-Gaussian with parameter

$$\sigma_m = \frac{2}{\sqrt{3}}\frac{2^m}{n}\sqrt{\|\mathbf{k}_{\text{split}}\|_\infty \log(\frac{6m}{2^m\delta^\star})},$$

that is,

$$\mathbb{E}[\exp(\langle \mathcal{W}_m, f\rangle_{\mathbf{k}_{\text{split}}})] \leq \exp(\tfrac{1}{2}\sigma_m^2\|f\|_{\mathbf{k}_{\text{split}}}^2) \quad \text{for all} \quad f \in \mathcal{H}_{\text{split}}, \tag{10}$$

where we note that the analysis of Dwivedi & Mackey (2021) remains unaffected when we replace $\|\mathbf{k}_{\text{split}}\|_\infty$ by $\|\mathbf{k}_{\text{split}}\|_{\infty,\text{in}}$ in all the arguments. Applying the sub-Gaussian Hoeffding inequality (Wainwright, 2019, Prop. 2.5) along with (10), we obtain that

$$\mathbb{P}[|\langle \mathcal{W}_m, f\rangle_{\mathbf{k}_{\text{split}}}| > t] \leq 2\exp(-\tfrac{1}{2}t^2/(\sigma_m^2\|f\|_{\mathbf{k}_{\text{split}}}^2)) \leq \delta' \text{ for } t \triangleq \sigma_m\|f\|_{\mathbf{k}_{\text{split}}}\sqrt{2\log(\frac{2}{\delta'})}.$$

Call this event $\mathcal{E}_{\text{sg}}$. As noted above, conditional to the event $\mathcal{E}_{\text{equi}}$, we also have

$$\mathcal{W}_m \stackrel{d}{=} \widetilde{\mathcal{W}}_m \quad \implies \quad \langle \mathcal{W}_m, f\rangle_{\mathbf{k}_{\text{split}}} \stackrel{d}{=} \mathbb{P}_{\text{in}}f - \mathbb{P}_{\text{out}}^{(1)}f,$$

where $\stackrel{d}{=}$ denotes equality in distribution. Furthermore, Dwivedi & Mackey (2021, Eqn. 48) implies that

$$\mathbb{P}(\mathcal{E}_{\text{equi}}) \geq 1 - \sum_{j=1}^m \frac{2^{j-1}}{m}\sum_{i=1}^{n/2^j}\delta_i.$$

Putting the pieces together, we have

$$\mathbb{P}[|\mathbb{P}_{\text{in}}f - \mathbb{P}_{\text{out}}^{(1)}f| \leq t] \geq \mathbb{P}(\mathcal{E}_{\text{equi}} \cap \mathcal{E}_{\text{sg}}^c) \geq \mathbb{P}(\mathcal{E}_{\text{equi}}) - \mathbb{P}(\mathcal{E}_{\text{sg}}) \geq 1 - \sum_{j=1}^m \frac{2^{j-1}}{m}\sum_{i=1}^{n/2^j}\delta_i - \delta' = p_{\text{sg}},$$

as claimed. The proof is now complete.

## C    Proof of Cor. 1: Guarantees for functions outside of $\mathcal{H}_{\text{SPLIT}}$

Fix any index $\ell \in [2^m]$, scalar $\delta' \in (0,1)$, and $f$ defined on $\mathcal{S}_{\text{in}}$, and consider the associated vector $g \in \mathbb{R}^n$ with $g_i = f(x_i)$ for each $i \in [n]$. We define two extended functions by appending the domain by $\mathbb{R}^n$ as follows: For any $w \in \mathbb{R}^n$, define $f_1((x,w)) = f(x)$ and $f_2((x,w)) = \langle g, w\rangle$

(the Euclidean inner product). Then we note that these functions replicate the values of $f$ on $\mathcal{S}_{\text{in}}$, since $f_1((x_i, w)) = f(x_i)$ for arbitrary $w \in \mathbb{R}^n$, and $f_2((x_i, e_i)) = \langle g, e_i \rangle = g_i = f(x_i)$, where $e_i$ denotes the $i$-th basis vector in $\mathbb{R}^n$. Thus we can write

$$\mathbb{P}_{\text{in}} f - \mathbb{P}_{\text{split}}^{(\ell)} f = \mathbb{P}'_{\text{in}} f_1 - \mathbb{P}'^{(\ell)}_{\text{split}} f_1 = \mathbb{P}'_{\text{in}} f_2 - \mathbb{P}'^{(\ell)}_{\text{split}} f_2 \tag{11}$$

for the extended empirical distributions $\mathbb{P}'_{\text{in}} = \frac{1}{n} \sum_{i=1}^n \delta_{x_i, e_i}$ and $\mathbb{P}'^{(\ell)}_{\text{split}}$, defined analogously. Notably, any function of the form $\tilde{f}((x, w)) = \langle \tilde{g}, w \rangle$ belongs to the RKHS of $\mathbf{k}'_{\text{split}}$ with

$$\|\tilde{f}\|_{\mathbf{k}'_{\text{split}}} \leq \|\tilde{g}\|_2 \tag{12}$$

by Berlinet & Thomas-Agnan (2011, Thm. 5).

By the repeated halving interpretation of kernel thinning (Dwivedi & Mackey, 2021, Sec. 5.2), on an event $\mathcal{E}$ of probability at least $p_{\text{sg}} + \delta'$ we may write

$$\mathbb{P}'_{\text{in}} f_2 - \mathbb{P}'^{(\ell)}_{\text{split}} f_2 = \sum_{j=1}^m \langle \mathcal{W}_j, f_2 \rangle_{\mathbf{k}'_{\text{split}}} = \sum_{j=1}^m \langle \mathcal{W}_j, f_{2,j} \rangle_{\mathbf{k}'_{\text{split}}}$$

where $\mathcal{W}_j$ denotes suitable random functions in the RKHS of $\mathbf{k}'_{\text{split}}$, and each $f_{2,j}((x, w)) = \langle g^{(j)}, w \rangle$ for $g^{(j)} \in \mathbb{R}^n$ a sparsification of $g$ with at most $\frac{n}{2^{j-1}}$ non-zero entries that satisfy

$$\mathbb{E}[\exp(\langle \mathcal{W}_j, f_{2,j} \rangle_{\mathbf{k}'_{\text{split}}}) \mid \mathcal{W}_{j-1}] \leq \exp(\frac{\sigma_j^2}{2} \|f_{2,j}\|^2_{\mathbf{k}'_{\text{split}}}) \overset{(12)}{\leq} \exp(\frac{\sigma_j^2}{2} \|g^{(j)}\|^2_2) \leq \exp(\frac{\sigma_j^2}{2} \frac{n}{2^{j-1}} \|f\|^2_{\infty, \text{in}})$$

for $\mathcal{W}_0 \triangleq 0$ and

$$\sigma_j^2 = 4(\tfrac{2^{j-1}}{n})^2 \|\mathbf{k}'_{\text{split}}\|_{\infty, \text{in}} \log(\tfrac{4m}{2^j \delta^\star}) \leq 2 \cdot \tfrac{4^j}{n^2} \log(\tfrac{4m}{2^j \delta^\star}),$$

since by definition $\|\mathbf{k}'_{\text{split}}\|_{\infty, \text{in}} \leq 2$. Hence, by sub-Gaussian additivity (see, e.g., Dwivedi & Mackey, 2021, Lem. 8), $\mathbb{P}_{\text{in}} f_2 - \mathbb{P}_{\text{split}}^{(\ell)} f_2$ is $\widetilde{\sigma}_2$ sub-Gaussian with

$$\begin{aligned}
\widetilde{\sigma}_2^2 &\leq \tfrac{4}{n} \|f\|^2_{\infty, \text{in}} \cdot \sum_{j=1}^m 2^j \log(\tfrac{4m}{2^j \delta^\star}) \overset{(i)}{=} \tfrac{4}{n} \|f\|^2_{\infty, \text{in}} \cdot 2\big((2^m - 1) \log(\tfrac{4m}{\delta^\star}) - ((2^m - 1)(m-1) + m) \log 2\big) \\
&= \tfrac{4}{n} \|f\|^2_{\infty, \text{in}} \cdot 2\big((2^m - 1) \log(\tfrac{4m \cdot 2}{2^m \delta^\star}) - m \log 2\big) \\
&\leq 8 \cdot \tfrac{2^m}{n} \|f\|^2_{\infty, \text{in}} \cdot \log(\tfrac{8m}{2^m \delta^\star}),
\end{aligned}$$

i.e.,

$$\widetilde{\sigma}_2 \leq \sqrt{\tfrac{2^m}{n}} \cdot \|f\|_{\infty, \text{in}} \cdot \sqrt{8 \log(\tfrac{8m}{2^m \delta^\star})} \tag{13}$$

on the event $\mathcal{E}$, where step (i) makes use of the following expressions:

$$\sum_{j=1}^m 2^j = 2(2^m - 1) \quad \text{and} \quad \sum_{j=1}^m j 2^j = 2((m-1)(2^m - 1) + m).$$

Moreover, when $f \in \mathcal{H}_{\text{split}}$, we additionally have $f_1$ in the RKHS of $\mathbf{k}'_{\text{split}}$ with

$$\|f_1\|_{\mathbf{k}'_{\text{split}}} \leq \|f\|_{\mathbf{k}_{\text{split}}} \sqrt{\|\mathbf{k}_{\text{split}}\|_\infty},$$

as argued in the proof of (23). The proof of Thm. 1 then implies that $\mathbb{P}'_{\text{in}} f_1 - \mathbb{P}'^{(\ell)}_{\text{split}} f_1$ is $\widetilde{\sigma}_1$ sub-Gaussian with

$$\widetilde{\sigma}_1 \leq \|f_a\|_{\mathbf{k}'_{\text{split}}} \tfrac{2}{\sqrt{3}} \tfrac{2^m}{n} \sqrt{\|\mathbf{k}'_{\text{split}}\|_{\infty, \text{in}} \cdot \log(\tfrac{6m}{2^m \delta^\star})} \leq \tfrac{2^m}{n} \cdot \|f\|_{\mathbf{k}_{\text{split}}} \sqrt{\|\mathbf{k}_{\text{split}}\|_\infty} \cdot \sqrt{\tfrac{8}{3} \log(\tfrac{6m}{2^m \delta^\star})}, \tag{14}$$

on the very same event $\mathcal{E}$.

Recalling (11) and putting the pieces together with the definitions (13) and (14), we conclude that on the event $\mathcal{E}$, the random variable $\mathbb{P}_{\text{in}} f - \mathbb{P}_{\text{split}}^{(\ell)} f$ is $\widetilde{\sigma}$ sub-Gaussian for

$$\widetilde{\sigma} \triangleq \min(\widetilde{\sigma}_1, \widetilde{\sigma}_2) \overset{(13),(14)}{\leq} \min\big(\sqrt{\tfrac{n}{2^m}} \|f\|_{\infty, \text{in}}, \|f\|_{\mathbf{k}_{\text{split}}} \sqrt{\|\mathbf{k}_{\text{split}}\|_\infty}\big) \cdot \tfrac{2^m}{n} \sqrt{8 \log(\tfrac{8m}{2^m \delta^\star})}.$$

The advertised high-probability bound (3) now follows from the $\widetilde{\sigma}$ sub-Gaussianity on $\mathcal{E}$ exactly as in the proof of Thm. 1.

## D  PROOF OF THM. 2: MMD GUARANTEE FOR TARGET KT

First, we note that by design, KT-SWAP ensures
$$\mathrm{MMD}_{\mathbf{k}}(\mathcal{S}_{\mathrm{in}}, \mathcal{S}_{\mathrm{KT}}) \leq \mathrm{MMD}_{\mathbf{k}}(\mathcal{S}_{\mathrm{in}}, \mathcal{S}^{(m,1)}),$$
where $\mathcal{S}^{(m,1)}$ denotes the first coreset returned by KT-SPLIT. Thus it suffices to show that $\mathrm{MMD}_{\mathbf{k}}(\mathcal{S}_{\mathrm{in}}, \mathcal{S}^{(m,1)})$ is bounded by the term stated on the right hand side of (5). Let $\mathbb{P}_{\mathrm{out}}^{(1)} \triangleq \frac{1}{n/2^m} \sum_{x \in \mathcal{S}^{(m,1)}} \boldsymbol{\delta}_x$. By design of KT-SPLIT, $\mathrm{supp}(\mathbb{P}_{\mathrm{out}}^{(1)}) \subseteq \mathrm{supp}(\mathbb{P}_{\mathrm{in}})$. Recall the set $\mathcal{A}$ is such that $\mathrm{supp}(\mathbb{P}_{\mathrm{in}}) \subseteq \mathcal{A}$.

**Proof of (5)**  Let $\mathcal{C} \triangleq \mathcal{C}_{\mathbf{k},\varepsilon}(\mathcal{A})$ denote the cover of minimum cardinality satisfying (4). Fix any $f \in \mathcal{B}_{\mathbf{k}}$. By the triangle inequality and the covering property (4) of $\mathcal{C}$, we have

$$
\begin{aligned}
\left|(\mathbb{P}_{\mathrm{in}} - \mathbb{P}_{\mathrm{out}}^{(1)})f\right| &\leq \inf_{g \in \mathcal{C}} \left|(\mathbb{P}_{\mathrm{in}} - \mathbb{P}_{\mathrm{out}}^{(1)})(f - g)\right| + \left|(\mathbb{P}_{\mathrm{in}} - \mathbb{P}_{\mathrm{out}}^{(1)})(g)\right| \\
&\leq \inf_{g \in \mathcal{C}} |\mathbb{P}_{\mathrm{in}}(f - g)| + \left|\mathbb{P}_{\mathrm{out}}^{(1)}(f - g)\right| + \sup_{g \in \mathcal{C}} \left|(\mathbb{P}_{\mathrm{in}} - \mathbb{P}_{\mathrm{out}}^{(1)})(g)\right| \\
&\leq \inf_{g \in \mathcal{C}} 2 \sup_{x \in \mathcal{A}} |f(x) - g(x)| + \sup_{g \in \mathcal{C}} \left|(\mathbb{P}_{\mathrm{in}} - \mathbb{P}_{\mathrm{out}}^{(1)})(g)\right| \\
&\leq 2\varepsilon + \sup_{g \in \mathcal{C}} \left|(\mathbb{P}_{\mathrm{in}} - \mathbb{P}_{\mathrm{out}}^{(1)})(g)\right|.
\end{aligned}
\tag{15}
$$

Applying Thm. 1, we have
$$\left|(\mathbb{P}_{\mathrm{in}} - \mathbb{P}_{\mathrm{out}}^{(1)})(g)\right| \leq \frac{2^m}{n} \|g\|_{\mathbf{k}} \sqrt{\frac{8}{3} \|\mathbf{k}\|_{\infty,\mathrm{in}} \cdot \log(\frac{4}{\delta^\star}) \log(\frac{4}{\delta'})} \tag{16}$$

with probability at least $1 - \delta' - \sum_{j=1}^{m} \frac{2^{j-1}}{m} \sum_{i=1}^{n/2^j} \delta_i = p_{\mathrm{sg}} - \delta'$. A standard union bound then yields that

$$\sup_{g \in \mathcal{C}} \left|(\mathbb{P}_{\mathrm{in}} - \mathbb{P}_{\mathrm{out}}^{(1)})(g)\right| \leq \frac{2^m}{n} \sup_{g \in \mathcal{C}} \|g\|_{\mathbf{k}} \sqrt{\frac{8}{3} \|\mathbf{k}\|_{\infty,\mathrm{in}} \cdot \log(\frac{4}{\delta^\star}) \left[\log |\mathcal{C}| + \log(\frac{4}{\delta'})\right]}$$

probability at least $p_{\mathrm{sg}} - \delta'$. Since $f \in \mathcal{B}_{\mathbf{k}}$ was arbitrary, and $\mathcal{C} \subset \mathcal{B}_{\mathbf{k}}$ and thus $\sup_{g \in \mathcal{C}} \|g\|_{\mathbf{k}} \leq 1$, we therefore have

$$
\begin{aligned}
\mathrm{MMD}_{\mathbf{k}}(\mathcal{S}_{\mathrm{in}}, \mathcal{S}^{(m,1)}) = \sup_{\|f\|_{\mathbf{k}} \leq 1} \left|(\mathbb{P}_{\mathrm{in}} - \mathbb{P}_{\mathrm{out}}^{(1)})f\right| &\overset{(15)}{\leq} 2\varepsilon + \sup_{g \in \mathcal{C}} \left|(\mathbb{P}_{\mathrm{in}} - \mathbb{P}_{\mathrm{out}}^{(1)})(g)\right| \\
&\leq 2\varepsilon + \sqrt{\frac{8\|\mathbf{k}\|_\infty}{3}} \cdot \frac{2^m}{n} \sqrt{\log(\frac{4}{\delta^\star}) \left[\log |\mathcal{C}| + \log(\frac{4}{\delta'})\right]},
\end{aligned}
$$

with probability at least $p_{\mathrm{sg}} - \delta'$ as claimed.

## E  PROOF OF THM. 3: MMD GUARANTEE FOR POWER KT

**Definition of $\widetilde{\mathfrak{M}}_\alpha$ and $\mathfrak{R}_{\max}$**  Define the $\mathbf{k}_\alpha$ tail radii,

$$\mathfrak{R}_{\mathbf{k}_\alpha, n}^\dagger \triangleq \min\{r : \tau_{\mathbf{k}_\alpha}(r) \leq \frac{\|\mathbf{k}_\alpha\|_\infty}{n}\}, \quad \text{where} \quad \tau_{\mathbf{k}_\alpha}(R) \triangleq \left(\sup_x \int_{\|y\|_2 \geq R} \mathbf{k}_\alpha^2(x, x - y) dy\right)^{\frac{1}{2}},$$

$$\mathfrak{R}_{\mathbf{k}_\alpha, n} \triangleq \min\{r : \sup_{\|x-y\|_2 \geq r} |\mathbf{k}_\alpha(x, y)| \leq \frac{\|\mathbf{k}_\alpha\|_\infty}{n}\}, \tag{17}$$

and the $\mathcal{S}_{\mathrm{in}}$ tail radii

$$\mathfrak{R}_{\mathcal{S}_{\mathrm{in}}} \triangleq \max_{x \in \mathcal{S}_{\mathrm{in}}} \|x\|_2, \quad \text{and} \quad \mathfrak{R}_{\mathcal{S}_{\mathrm{in}}, \mathbf{k}_\alpha, n} \triangleq \min\left(\mathfrak{R}_{\mathcal{S}_{\mathrm{in}}}, n^{1+\frac{1}{d}} \mathfrak{R}_{\mathbf{k}_\alpha, n} + n^{\frac{1}{d}} \|\mathbf{k}_\alpha\|_\infty / L_{\mathbf{k}_\alpha}\right). \tag{18}$$

Furthermore, define the inflation factor

$$\mathfrak{M}_{\mathbf{k}_\alpha}(n, m, d, \delta, \delta', R) \triangleq 37 \sqrt{\log\left(\frac{6m}{2^m \delta}\right)} \left[\sqrt{\log\left(\frac{4}{\delta'}\right)} + 5\sqrt{d \log(2 + 2\frac{L_{\mathbf{k}_\alpha}}{\|\mathbf{k}_\alpha\|_\infty}(\mathfrak{R}_{\mathbf{k}_\alpha, n} + R))}\right],$$

where $L_{\mathbf{k}_\alpha}$ denotes a Lipschitz constant satisfying $|\mathbf{k}_\alpha(x, y) - \mathbf{k}_\alpha(x, z)| \leq L_{\mathbf{k}_\alpha} \|y - z\|_2$ for all $x, y, z \in \mathbb{R}^d$. With the notations in place, we can define the quantities appearing in Thm. 3:

$$\widetilde{\mathfrak{M}}_\alpha \triangleq \mathfrak{M}_{\mathbf{k}_\alpha}(n, m, d, \delta^\star, \delta', \mathfrak{R}_{\mathcal{S}_{\mathrm{in}}, \mathbf{k}_\alpha, n}) \quad \text{and} \quad \mathfrak{R}_{\max} \triangleq \max(\mathfrak{R}_{\mathcal{S}_{\mathrm{in}}}, \mathfrak{R}_{\mathbf{k}_\alpha, n/2^m}^\dagger). \tag{19}$$

The scaling of these two parameters depends on the tail behavior of $\mathbf{k}_\alpha$ and the growth of the radii $\mathfrak{R}_{\mathcal{S}_{\mathrm{in}}}$ (which in turn would typically depend on the tail behavior of $\mathbb{P}$). The scaling of $\widetilde{\mathfrak{M}}_\alpha$ and $\mathfrak{R}_{\max}$ stated in Thm. 3 under the compactly supported or subexponential tail conditions follows directly from Dwivedi & Mackey (2021, Tab. 2, App. I).

**Proof of Thm. 3** The KT-SWAP step ensures that

$$\mathrm{MMD}_{\mathbf{k}}(\mathcal{S}_{\mathrm{in}}, \mathcal{S}_{\alpha\mathrm{KT}}) \leq \mathrm{MMD}_{\mathbf{k}}(\mathcal{S}_{\mathrm{in}}, \mathcal{S}_{\alpha}^{(m,1)}),$$

where $\mathcal{S}_{\alpha}^{(m,1)}$ denotes the first coreset output by KT-SPLIT with $\mathbf{k}_{\mathrm{split}} = \mathbf{k}_{\alpha}$. Next, we state a key interpolation result for $\mathrm{MMD}_{\mathbf{k}}$ that relates it to the MMD of its power kernels (Def. 2) (see App. G for the proof).

**Proposition 1 (An interpolation result for MMD)** *Consider a shift-invariant kernel* $\mathbf{k}$ *that admits valid* $\alpha$ *and* $2\alpha$-*power kernels* $\mathbf{k}_{\alpha}$ *and* $\mathbf{k}_{2\alpha}$ *respectively for some* $\alpha \in [\frac{1}{2}, 1]$. *Then for any two discrete measures* $\mathbb{P}$ *and* $\mathbb{Q}$ *supported on finitely many points, we have*

$$\mathrm{MMD}_{\mathbf{k}}(\mathbb{P}, \mathbb{Q}) \leq (\mathrm{MMD}_{\mathbf{k}_{\alpha}}(\mathbb{P}, \mathbb{Q}))^{2-\frac{1}{\alpha}} \cdot (\mathrm{MMD}_{\mathbf{k}_{2\alpha}}(\mathbb{P}, \mathbb{Q}))^{\frac{1}{\alpha}-1}. \tag{20}$$

Given Prop. 1, it remains to establish suitable upper bounds on MMDs of $\mathbf{k}_{\alpha}$ and $\mathbf{k}_{2\alpha}$. To this end, first we note that for any reproducing kernel $\mathbf{k}$ and any two distributions $\mathbb{P}$ and $\mathbb{Q}$, Hölder's inequality implies that

$$\mathrm{MMD}_{\mathbf{k}}^{2}(\mathbb{P}, \mathbb{Q}) = \|(\mathbb{P} - \mathbb{Q})\mathbf{k}\|_{\mathbf{k}}^{2} = (\mathbb{P} - \mathbb{Q})(\mathbb{P} - \mathbb{Q})\mathbf{k} \leq \|\mathbb{P} - \mathbb{Q}\|_{1}\|(\mathbb{P} - \mathbb{Q})\mathbf{k}\|_{\infty}$$
$$\leq 2\|(\mathbb{P} - \mathbb{Q})\mathbf{k}\|_{\infty}.$$

Now, let $\mathbb{P}_{\mathrm{in}}$ and $\mathbb{P}_{\alpha}^{(m,1)}$ denote the empirical distributions of $\mathcal{S}_{\mathrm{in}}$ and $\mathcal{S}_{\alpha}^{(m,1)}$. Now applying Dwivedi & Mackey (2021, Thm. 4(b)), we find that

$$\mathrm{MMD}_{\mathbf{k}_{\alpha}}(\mathcal{S}_{\mathrm{in}}, \mathcal{S}_{\alpha}^{(m,1)}) \leq \sqrt{2\|(\mathbb{P}_{\mathrm{in}} - \mathbb{P}_{\alpha}^{(m,1)})\mathbf{k}_{\alpha}\|_{\infty,\mathrm{in}}} \leq \sqrt{2 \cdot \frac{2^{m}}{n}\|\mathbf{k}_{\alpha}\|_{\infty,\mathrm{in}}\widetilde{\mathfrak{M}}_{\mathbf{k}_{\alpha}}} \tag{21}$$

with probability $p_{\mathrm{sg}} - \delta'$, where $\widetilde{\mathfrak{M}}_{\mathbf{k}_{\alpha}}$ was defined in (19). We note that while Dwivedi & Mackey (2021, Thm. 4(b)) uses $\|\mathbf{k}_{\alpha}\|_{\infty}$ in their bounds, we can replace it by $\|\mathbf{k}_{\alpha}\|_{\infty,\mathrm{in}}$, and verifying that all the steps of the proof continue to be valid (noting that $\|\mathbf{k}_{\alpha}\|_{\infty,\mathrm{in}}$ is deterministic given $\mathcal{S}_{\mathrm{in}}$). Furthermore, Dwivedi & Mackey (2021, Thm. 4(b)) yields that

$$\mathrm{MMD}_{\mathbf{k}_{2\alpha}}(\mathcal{S}_{\mathrm{in}}, \mathcal{S}_{\alpha}^{(m,1)}) \leq \frac{2^{m}}{n}\|\mathbf{k}_{\alpha}\|_{\infty,\mathrm{in}}\left(2 + \sqrt{\frac{(4\pi)^{d/2}}{\Gamma(\frac{d}{2}+1)} \cdot \mathfrak{R}_{\max}^{\frac{d}{2}} \cdot \widetilde{\mathfrak{M}}_{\alpha}}\right), \tag{22}$$

with probability $p_{\mathrm{sg}} - \delta'$, where we have once again replaced the term $\|\mathbf{k}_{\alpha}\|_{\infty}$ with $\|\mathbf{k}_{\alpha}\|_{\infty,\mathrm{in}}$ for the same reasons as stated above. We note that the two bounds (21) and (22) apply under the same high probability event as noted in Dwivedi & Mackey (2021, proof of Thm. 1, eqn. (18)). Putting together the pieces, we find that

$$\mathrm{MMD}_{\mathbf{k}}(\mathcal{S}_{\mathrm{in}}, \mathcal{S}_{\alpha}^{(m,1)}) \overset{(20)}{\leq} (\mathrm{MMD}_{\mathbf{k}_{\alpha}}(\mathcal{S}_{\mathrm{in}}, \mathcal{S}_{\alpha}^{(m,1)})^{2-\frac{1}{\alpha}} \cdot (\mathrm{MMD}_{\mathbf{k}_{2\alpha}}(\mathcal{S}_{\mathrm{in}}, \mathcal{S}_{\alpha}^{(m,1)}))^{\frac{1}{\alpha}-1}$$

$$\overset{(21,22)}{\leq} \left[2 \cdot \frac{2^{m}}{n}\|\mathbf{k}_{\alpha}\|_{\infty,\mathrm{in}}\widetilde{\mathfrak{M}}_{\alpha}\right]^{1-\frac{1}{2\alpha}}\left[\frac{2^{m}}{n}\|\mathbf{k}_{\alpha}\|_{\infty,\mathrm{in}}\left(2 + \sqrt{\frac{(4\pi)^{d/2}}{\Gamma(\frac{d}{2}+1)} \cdot \mathfrak{R}_{\max}^{\frac{d}{2}} \cdot \widetilde{\mathfrak{M}}_{\alpha}}\right)\right]^{\frac{1}{\alpha}-1}$$

$$= \left(\frac{2^{m}}{n}\|\mathbf{k}_{\alpha}\|_{\infty,\mathrm{in}}\right)^{\frac{1}{2\alpha}}(2 \cdot \widetilde{\mathfrak{M}}_{\alpha})^{1-\frac{1}{2\alpha}}\left(2 + \sqrt{\frac{(4\pi)^{d/2}}{\Gamma(\frac{d}{2}+1)} \cdot \mathfrak{R}_{\max}^{\frac{d}{2}} \cdot \widetilde{\mathfrak{M}}_{\alpha}}\right)^{\frac{1}{\alpha}-1},$$

as claimed. The proof is now complete.

## F  PROOF OF THM. 4: SINGLE FUNCTION & MMD GUARANTEES FOR KT+

**Proof of (8)** First, we note that the RKHS $\mathcal{H}$ of $\mathbf{k}$ is contained in the RKHS $\mathcal{H}^{\dagger}$ of $\mathbf{k}^{\dagger}$ Berlinet & Thomas-Agnan (2011, Thm. 5). Now, applying Thm. 1 with $\mathbf{k}_{\mathrm{split}} = \mathbf{k}^{\dagger}$ for any fixed function $f \in \mathcal{H} \subset \mathcal{H}^{\dagger}$ and $\delta' \in (0, 1)$, we obtain that

$$\left|\mathbb{P}_{\mathrm{in}}f - \mathbb{P}_{\mathrm{split}}^{(\ell)}f\right| \leq \|f\|_{\mathbf{k}^{\dagger}} \cdot \frac{2}{\sqrt{3}}\frac{2^{m}}{n}\sqrt{\|\mathbf{k}^{\dagger}\|_{\infty,\mathrm{in}} \cdot \log(\frac{6m}{2^{m}\delta^{\star}})}\sqrt{2\log(\frac{2}{\delta'})}$$

$$\overset{(i)}{\leq} \|f\|_{\mathbf{k}^{\dagger}} \cdot \frac{2^{m}}{n}\sqrt{\frac{16}{3}\log(\frac{6m}{2^{m}\delta^{\star}})\log(\frac{2}{\delta'})},$$

$$\overset{(ii)}{\leq} \|f\|_{\mathbf{k}} \cdot \frac{2^{m}}{n}\sqrt{\frac{16}{3}\|\mathbf{k}\|_{\infty}\log(\frac{6m}{2^{m}\delta^{\star}})\log(\frac{2}{\delta'})},$$

with probability at least $p_{\text{sg}}$. Here step (i) follows from the inequality $\|\mathbf{k}^\dagger\|_\infty \leq 2$, and step (ii) follows from the inequality $\|f\|_{\mathbf{k}^\dagger} \leq \sqrt{\|\mathbf{k}\|_\infty}\|f\|_{\mathbf{k}}$, which in turn follows from the standard facts that

$$\|f\|_{\lambda\mathbf{k}} \stackrel{(iii)}{=} \frac{\|f\|_{\mathbf{k}}}{\sqrt{\lambda}}, \quad \text{and} \quad \|f\|_{\lambda\mathbf{k}+\mathbf{k}_\alpha} \stackrel{(iv)}{\leq} \|f\|_{\lambda\mathbf{k}} \quad \text{for} \quad \lambda > 0, f \in \mathcal{H}, \tag{23}$$

see, e.g., Zhang & Zhao (2013, Proof of Prop. 2.5) for a proof of step (iii), Berlinet & Thomas-Agnan (2011, Thm. 5) for step (iv). The proof for the bound (8) is now complete. $\qquad\square$

**Proof of (9)** Repeating the proof of Thm. 2 with the bound (16) replaced by (8) yields that

$$\text{MMD}_{\mathbf{k}}(\mathcal{S}_{\text{in}}, \mathcal{S}_{\text{KT+}}) \leq \inf_{\varepsilon, \mathcal{S}_{\text{in}} \subset \mathcal{A}} 2\varepsilon + \frac{2^m}{n}\sqrt{\frac{16}{3}\|\mathbf{k}\|_\infty \log(\frac{6m}{2^m\delta^\star}) \cdot \left[\log(\frac{4}{\delta'}) + \mathcal{M}_{\mathbf{k}}(\mathcal{A}, \varepsilon)\right]}$$
$$\leq \sqrt{2} \cdot \overline{\mathbf{M}}_{\text{targetKT}}(\mathbf{k}) \tag{24}$$

with probability at least $p_{\text{sg}}$. Let us denote this event by $\mathcal{E}_1$.

To establish the other bound, first we note that KT-SWAP step ensures that

$$\text{MMD}_{\mathbf{k}}(\mathcal{S}_{\text{in}}, \mathcal{S}_{\text{KT+}}) \leq \text{MMD}_{\mathbf{k}}(\mathcal{S}_{\text{in}}, \mathcal{S}_{\text{KT+}}^{(m,1)}), \tag{25}$$

where $\mathcal{S}_{\text{KT+}}^{(m,1)}$ denotes the first coreset output by KT-SPLIT with $\mathbf{k}_{\text{split}} = \mathbf{k}^\dagger$. We can now repeat the proof of Thm. 3, using the sub-Gaussian tail bound (8), and with a minor substitution, namely, $\|\mathbf{k}_\alpha\|_{\infty,\text{in}}$ replaced by $2\|\mathbf{k}_\alpha\|_\infty$. Putting it together with (25), we conclude that

$$\text{MMD}_{\mathbf{k}}(\mathcal{S}_{\text{in}}, \mathcal{S}_{\text{KT+}}) \leq \left(\frac{2^m}{n}2\|\mathbf{k}_\alpha\|_{\infty,\text{in}}\right)^{\frac{1}{2\alpha}}(2\widetilde{\mathfrak{M}}_\alpha)^{1-\frac{1}{2\alpha}}\left(2 + \sqrt{\frac{(4\pi)^{d/2}}{\Gamma(\frac{d}{2}+1)} \cdot \mathfrak{R}_{\max}^{\frac{d}{2}} \cdot \widetilde{\mathfrak{M}}_\alpha}\right)^{\frac{1}{\alpha}-1}$$
$$= 2^{\frac{1}{2\alpha}} \cdot \overline{\mathbf{M}}_{\text{powerKT}}(\mathbf{k}_\alpha), \tag{26}$$

with probability at least $p_{\text{sg}}$. Let us denote this event by $\mathcal{E}_2$.

Note that the quantities on the right hand side of the bounds (24) and (26) are deterministic given $\mathcal{S}_{\text{in}}$, and thus can be computed apriori. Consequently, we apply the high probability bound only for one of the two events $\mathcal{E}_1$ or $\mathcal{E}_2$ depending on which of the two quantities (deterministically) attains the minimum. Thus, the bound (9) holds with probability at least $p_{\text{sg}}$ as claimed. $\qquad\square$

# G    PROOF OF PROP. 1: AN INTERPOLATION RESULT FOR MMD

For two arbitrary distributions $\mathbb{P}$ and $\mathbb{Q}$, and any reproducing kernel $\mathbf{k}$, Gretton et al. (2012, Lem. 4) yields that

$$\text{MMD}_{\mathbf{k}}^2(\mathbb{P}, \mathbb{Q}) = \|(\mathbb{P} - \mathbb{Q})\mathbf{k}\|_{\mathbf{k}}^2. \tag{27}$$

Let $\mathcal{F}$ denote the generalized Fourier transform (GFT) operator (Wendland (2004, Def. 8.9)). Since $\mathbf{k}(x, y) = \kappa(x - y)$, Wendland (2004, Thm. 10.21) yields that

$$\|f\|_{\mathbf{k}}^2 = \frac{1}{(2\pi)^{d/2}}\int_{\mathbb{R}^d}\frac{(\mathcal{F}(f)(\omega))^2}{\mathcal{F}(\kappa)(\omega)}d\omega, \quad \text{for} \quad f \in \mathcal{H}. \tag{28}$$

Let $\widehat{\kappa} \triangleq \mathcal{F}(\kappa)$, and consider a discrete measure $\mathbb{D} = \sum_{i=1}^n w_i\boldsymbol{\delta}_{x_i}$ supported on finitely many points, and let $\mathbb{D}\mathbf{k}(x) \triangleq \sum w_i\mathbf{k}(x, x_i) = \sum w_i\kappa(x - x_i)$. Now using the linearity of the GFT operator $\mathcal{F}$, we find that for any $\omega \in \mathbb{R}^d$,

$$\mathcal{F}(\mathbb{D}\mathbf{k})(\omega) = \mathcal{F}(\sum_{i=1}^n w_i\kappa(\cdot - x_i))(\omega) = \sum_{i=1}^n w_i\mathcal{F}(\kappa(\cdot - x_i)(\omega) = (\sum_{i=1}^n w_i e^{-\langle\omega, x_i\rangle}) \cdot \widehat{\kappa}(\omega)$$
$$= \widehat{D}(\omega)\widehat{\kappa}(\omega) \tag{29}$$

where we used the time-shifting property of GFT that $\mathcal{F}(\kappa(\cdot - x_i))(\omega) = e^{-\langle\omega, x_i\rangle}\widehat{\kappa}(\omega)$ (proven for completeness in Lem. 1), and used the shorthand $\widehat{D}(\omega) \triangleq (\sum_{i=1}^n w_i e^{-\langle\omega, x_i\rangle})$ in the last step.

Putting together (27) to (29) with $\mathbb{D} = \mathbb{P} - \mathbb{Q}$, we find that

$$
\begin{aligned}
\mathrm{MMD}_{\mathbf{k}}^2(\mathbb{P}, \mathbb{Q}) &= \frac{1}{(2\pi)^{d/2}} \int_{\mathbb{R}^d} \widehat{D}^2(\omega) \widehat{\kappa}(\omega) d\omega \qquad\qquad\qquad (30) \\
&= \frac{1}{(2\pi)^{d/2}} \int_{\mathbb{R}^d} \widehat{D}^2(\omega) \widehat{\kappa}^\alpha(\omega) (\widehat{\kappa}^\alpha(\omega))^{\frac{1-\alpha}{\alpha}} d\omega \\
&= \frac{1}{(2\pi)^{d/2}} \int_{\mathbb{R}^d} \widehat{D}^2(\omega') \widehat{\kappa}^\alpha(\omega') d\omega' \int_{\mathbb{R}^d} \frac{\widehat{D}^2(\omega)\widehat{\kappa}^\alpha(\omega)}{\int_{\mathbb{R}^d} \widehat{D}^2(\omega')\widehat{\kappa}^\alpha(\omega')d\omega'} (\widehat{\kappa}^\alpha(\omega))^{\frac{1-\alpha}{\alpha}} d\omega \\
&\overset{(i)}{\leq} \frac{1}{(2\pi)^{d/2}} \int_{\mathbb{R}^d} \widehat{D}^2(\omega') \widehat{\kappa}^\alpha(\omega') d\omega' \left( \int_{\mathbb{R}^d} \frac{\widehat{D}^2(\omega)\widehat{\kappa}^\alpha(\omega)}{\int_{\mathbb{R}^d} \widehat{D}^2(\omega')\widehat{\kappa}^\alpha(\omega')d\omega'} \widehat{\kappa}^\alpha(\omega)d\omega \right)^{\frac{1-\alpha}{\alpha}} \\
&= \frac{1}{(2\pi)^{d/2}} \left( \int_{\mathbb{R}^d} \widehat{D}^2(\omega') \widehat{\kappa}^\alpha(\omega') d\omega' \right)^{2-\frac{1}{\alpha}} \left( \int_{\mathbb{R}^d} \frac{\widehat{D}^2(\omega)\widehat{\kappa}^{2\alpha}(\omega)}{d} \omega \right)^{\frac{1-\alpha}{\alpha}} \\
&= \left( \frac{1}{(2\pi)^{d/2}} \int_{\mathbb{R}^d} \widehat{D}^2(\omega') \widehat{\kappa}^\alpha(\omega') d\omega' \right)^{2-\frac{1}{\alpha}} \left( \frac{1}{(2\pi)^{d/2}} \int_{\mathbb{R}^d} \frac{\widehat{D}^2(\omega)\widehat{\kappa}^{2\alpha}(\omega)}{d} \omega \right)^{\frac{1-\alpha}{\alpha}} \\
&\overset{(ii)}{=} (\mathrm{MMD}_{\mathbf{k}_\alpha}^2(\mathbb{P}, \mathbb{Q}))^{2-\frac{1}{\alpha}} \cdot (\mathrm{MMD}_{\mathbf{k}_{2\alpha}}^2(\mathbb{P}, \mathbb{Q}))^{\frac{1}{\alpha}-1},
\end{aligned}
$$

where step (i) makes use of Jensen's inequality and the fact that the function $t \mapsto t^{\frac{1-\alpha}{\alpha}}$ for $t \geq 0$ is concave for $\alpha \in [\frac{1}{2}, 1]$, and step (ii) follows by applying (30) for kernels $\mathbf{k}_\alpha$ and $\mathbf{k}_{2\alpha}$ and noting that by definition $\mathcal{F}(\mathbf{k}_\alpha) = \widehat{\kappa}^\alpha$, and $\mathcal{F}(\mathbf{k}_{2\alpha}) = \widehat{\kappa}^{2\alpha}$. Noting MMD is a non-negative quantity, and taking square-root establishes the claim (20).

**Lemma 1 (Shifting property of the generalized Fourier transform)** *If $\widehat{\kappa}$ denotes the generalized Fourier transform (GFT) (Wendland, 2004, Def. 8.9) of the function $\kappa : \mathbb{R}^d \to \mathbb{R}$, then $e^{-\langle \cdot, x_i \rangle} \widehat{\kappa}$ denotes the GFT of the shifted function $\kappa(\cdot - x_i)$, for any $x_i \in \mathbb{R}^d$.*

**Proof** Note that by definition of the GFT $\widehat{\kappa}$ (Wendland, 2004, Def. 8.9), we have

$$
\int \kappa(x) \widehat{\gamma}(x) dx = \int \widehat{\kappa}(\omega) \gamma(\omega) d\omega, \qquad\qquad (31)
$$

for all suitable Schwartz functions $\gamma$ (Wendland, 2004, Def. 5.17), where $\widehat{\gamma}$ denotes the Fourier transform (Wendland, 2004, Def. 5.15) of $\gamma$ since GFT and FT coincide for these functions (as noted in the discussion after Wendland (2004, Def. 8.9)). Thus to prove the lemma, we need to verify that

$$
\int \kappa(x - x_i) \widehat{\gamma}(x) dx = \int e^{-\langle \omega, x_i \rangle} \widehat{\kappa}(\omega) \gamma(\omega) d\omega, \qquad\qquad (32)
$$

for all suitable Schwartz functions $\gamma$. Starting with the right hand side of the display (32), we have

$$
\int e^{-\langle \omega, x_i \rangle} \widehat{\kappa}(\omega) \gamma(\omega) d\omega = \int \widehat{\kappa}(\omega)(e^{-\langle \omega, x_i \rangle} \gamma(\omega)) d\omega \overset{(i)}{=} \int \kappa(x) \widehat{\gamma}(x + x_i) dx \overset{(ii)}{=} \int \kappa(z - x_i) \widehat{\gamma}(z) dz,
$$

where step (i) follows from the shifting property of the FT (Wendland, 2004, Thm. 5.16(4)), and the fact that the GFT condition (31) holds for the shifted function $\gamma(\cdot + x_i)$ function as well since it is still a Schwartz function (recall that $\widehat{\gamma}$ is the FT), and step (ii) follows from a change of variable. We have thus established (32), and the proof is complete. $\qquad\square$

# H SUB-OPTIMALITY OF SINGLE FUNCTION GUARANTEES WITH ROOT KT

Define $\widetilde{\mathbf{k}}_{\mathrm{rt}}$ as the scaled version of $\mathbf{k}_{\mathrm{rt}}$, i.e., $\widetilde{\mathbf{k}}_{\mathrm{rt}} \triangleq \mathbf{k}_{\mathrm{rt}}/\|\mathbf{k}_{\mathrm{rt}}\|_\infty$ that is bounded by 1. Then Zhang & Zhao (2013, Proof of Prop. 2.3) implies that

$$
\|f\|_{\mathbf{k}_{\mathrm{rt}}} = \frac{1}{\sqrt{\|\mathbf{k}_{\mathrm{rt}}\|_\infty}} \|f\|_{\widetilde{\mathbf{k}}_{\mathrm{rt}}}. \qquad\qquad (33)
$$

And thus we also have $\mathcal{H}_{\mathrm{rt}} = \widetilde{\mathcal{H}}_{\mathrm{rt}}$ where $\mathcal{H}_{\mathrm{rt}}$ and $\widetilde{\mathcal{H}}_{\mathrm{rt}}$ respectively denote the RKHSs of $\mathbf{k}_{\mathrm{rt}}$ and $\widetilde{\mathbf{k}}_{\mathrm{rt}}$.

Next, we note that for any two kernels $\mathbf{k}_1$ and $\mathbf{k}_2$ with corresponding RKHSs $\mathcal{H}_1$ and $\mathcal{H}_2$ with $\mathcal{H}_1 \subset \mathcal{H}_2$, in the convention of Zhang & Zhao (2013, Lem. 2.2, Prop. 2.3), we have

$$
\frac{\|f\|_{\mathbf{k}_2}}{\|f\|_{\mathbf{k}_1}} \leq \beta(\mathcal{H}_1, \mathcal{H}_2) \leq \sqrt{\lambda(\mathcal{H}_1, \mathcal{H}_2)} \quad \text{for} \quad f \in \mathcal{H}. \qquad\qquad (34)
$$

Consequently, we have

$$\sqrt{\max_{x \in \mathcal{S}_{\mathrm{in}}} \mathbf{k}_{\mathrm{rt}}(x,x)} \frac{\|f\|_{\mathbf{k}_{\mathrm{rt}}}}{\|f\|_{\mathbf{k}}} \le \sqrt{\|\mathbf{k}_{\mathrm{rt}}\|_{\infty}} \frac{\|f\|_{\mathbf{k}_{\mathrm{rt}}}}{\|f\|_{\mathbf{k}}} \stackrel{(33)}{=} \frac{\|f\|_{\widetilde{\mathbf{k}}_{\mathrm{rt}}}}{\|f\|_{\mathbf{k}}} \le \sqrt{\lambda(\mathcal{H}, \widetilde{\mathcal{H}}_{\mathrm{rt}})}, \tag{35}$$

where in the last step, we have applied the bound (34) with $(\mathbf{k}_1, \mathcal{H}_1) \leftarrow (\mathbf{k}, \mathcal{H})$ and $(\mathbf{k}_2, \mathcal{H}_2) \leftarrow (\widetilde{\mathbf{k}}_{\mathrm{rt}}, \widetilde{\mathbf{k}}_{\mathrm{rt}})$ since $\mathcal{H} \subset \mathcal{H}_{\mathrm{rt}} = \widetilde{\mathbf{k}}_{\mathrm{rt}}$.

Next, we use (35) to the kernels studied in Dwivedi & Mackey (2021) where we note that all the kernels in that work were scaled to ensure $\|\mathbf{k}\|_{\infty} = 1$ and in fact satisfied $\mathbf{k}(x,x) = 1$. Consequently, the multiplicative factor stated in the discussion after Thm. 1, namely, $\sqrt{\frac{\|\mathbf{k}_{\mathrm{rt}}\|_{\infty,\mathrm{in}}}{\|\mathbf{k}\|_{\infty,\mathrm{in}}}} \frac{\|f\|_{\mathbf{k}_{\mathrm{rt}}}}{\|f\|_{\mathbf{k}}}$ can be bounded by $\sqrt{\lambda(\mathcal{H}, \widetilde{\mathcal{H}}_{\mathrm{rt}})}$ given the arguments above.

For $\mathbf{k} = \mathbf{Gauss}(\sigma)$ kernels, Zhang & Zhao (2013, Prop. 3.5(1)) yields that

$$\lambda(\mathcal{H}, \widetilde{\mathcal{H}}_{\mathrm{rt}}) = 2^{d/2}.$$

For $\mathbf{k} = \mathbf{B\text{-}spline}(2\beta + 1, \gamma)$ with $\beta \in 2\mathbb{N} + 1$, Zhang & Zhao (2013, Prop. 3.5(1)) yields that

$$\lambda(\mathcal{H}, \widetilde{\mathcal{H}}_{\mathrm{rt}}) = 1.$$

For $\mathbf{k} = \mathbf{Matérn}(\nu, \gamma)$ with $\nu > d$, some algebra along with Zhang & Zhao (2013, Prop 3.1) yields that

$$\lambda(\mathcal{H}, \widetilde{\mathcal{H}}_{\mathrm{rt}}) = \frac{\Gamma(\nu)\Gamma((\nu-d)/2)}{\Gamma(\nu-d/2)\Gamma(\nu/2)} \ge 1.$$

# I   ADDITIONAL EXPERIMENTAL RESULTS

This section provides additional experimental details and results deferred from Sec. 4.

**Common settings and error computation**   To obtain an output coreset of size $n^{\frac{1}{2}}$ with $n$ input points, we (a) take every $n^{\frac{1}{2}}$-th point for standard thinning (ST) and (b) run KT with $m = \frac{1}{2}\log_2 n$ using an ST coreset as the base coreset in KT-SWAP. For Gaussian and MoG target we use i.i.d. points as input, and for MCMC targets we use an ST coreset after burn-in as the input (see App. I for more details). We compute errors with respect to $\mathbb{P}$ whenever available in closed form and otherwise use $\mathbb{P}_{\mathrm{in}}$. For each input sample size $n \in \{2^4, 2^6, \dots, 2^{14}\}$ with $\delta_i = \frac{1}{2n}$, we report the mean MMD or function integration error $\pm 1$ standard error across 10 independent replications of the experiment (the standard errors are too small to be visible in all experiments). We also plot the ordinary least squares fit (for log mean error vs log coreset size), with the slope of the fit denoted as the empirical decay rate, e.g., for an OLS fit with slope $-0.25$, we display the decay rate of $n^{-0.25}$.

**Details of test functions**   We note the following: (a) For Gaussian targets, the error with CIF function and i.i.d. input is measured across the sample mean over the $n$ input points and $\sqrt{n}$ output points obtained by standard thinning the input sequence, since $\mathbb{P}f_{\mathrm{CIF}}$ does not admit a closed form. (b) To define the function $f : x \mapsto \mathbf{k}(X', x)$, first we draw a sample $X \sim \mathbb{P}$, independent of the input, and then set $X' = 2X$. For the MCMC targets, we draw a point uniformly from a held out data point not used as input for KT. For each target, the sample is drawn exactly once and then fixed throughout all sample sizes and repetions.

## I.1   MIXTURE OF GAUSSIANS EXPERIMENTS

Our mixture of Gaussians target is given by $\mathbb{P} = \frac{1}{M}\sum_{j=1}^{M} \mathcal{N}(\mu_j, \mathbf{I}_d)$ for $M \in \{4, 6, 8\}$ where

$$\mu_1 = [-3, 3]^\top, \quad \mu_2 = [-3, 3]^\top, \quad \mu_3 = [-3, -3]^\top, \quad \mu_4 = [3, -3]^\top,$$
$$\mu_5 = [0, 6]^\top, \quad \mu_6 = [-6, 0]^\top, \quad \mu_7 = [6, 0]^\top, \quad \mu_8 = [0, -6]^\top.$$

Two independent replicates of Fig. 1 can be found in Fig. 4. Finally, we display mean MMD ($\pm 1$ standard error across ten independent experiment replicates) as a function of coreset size in Fig. 5 for $M = 4, 6$ component MoG targets. The conclusions from Fig. 5 are identical to those from the bottom row of Fig. 1: TARGET KT and ROOT KT provide similar MMD errors with GAUSS $\mathbf{k}$, and all variants of KT provide a significant improvement over i.i.d. sampling both in terms of magnitude and decay rate with input size. Morever the observed decay rates for KT+ closely match the rates guaranteed by our theory in Tab. 3.

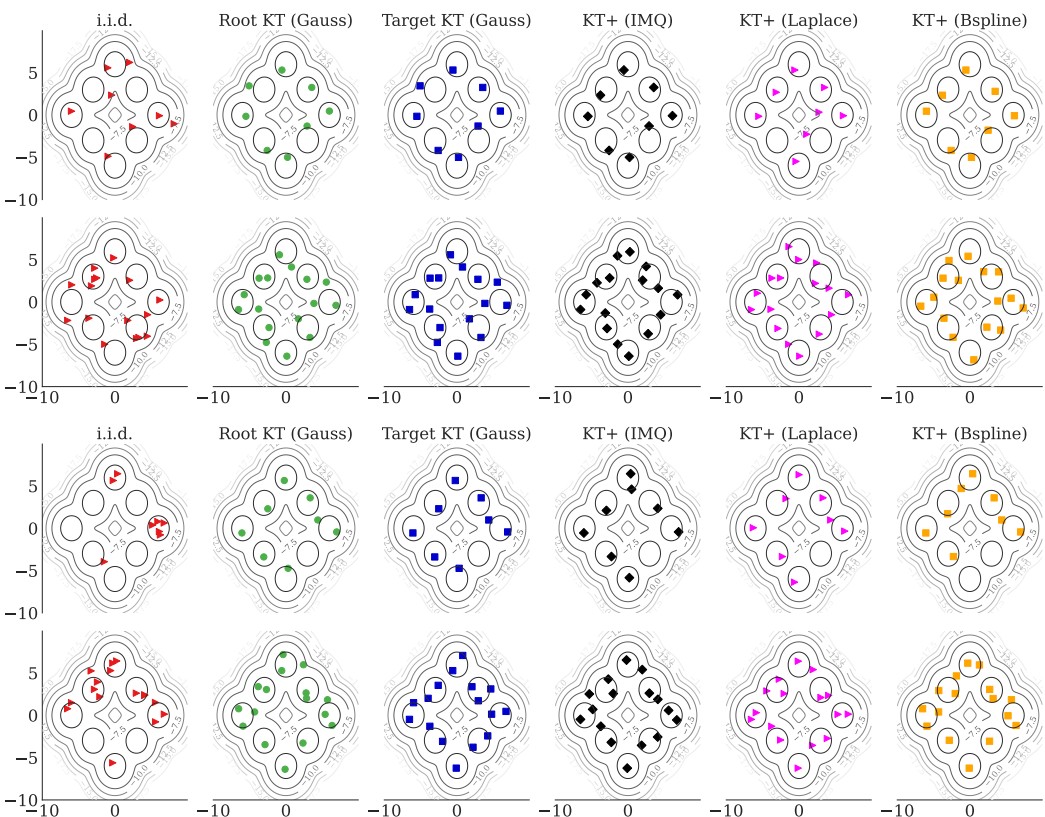

Figure 4: **Generalized kernel thinning (KT) and i.i.d. coresets** for various kernels **k** (in parentheses) and an 8-component mixture of Gaussian target $\mathbb{P}$ with equidensity contours underlaid. These plots are independent replicates of Fig. 1. See Sec. 4 for more details.

## I.2   MCMC EXPERIMENTS

Our set-up for MCMC experiments follows closely that of Dwivedi & Mackey (2021). For complete details on the targets and sampling algorithms we refer the reader to Riabiz et al. (2020a, Sec. 4).

**Goodwin and Lotka-Volterra experiments**   From Riabiz et al. (2020b), we use the output of four distinct MCMC procedures targeting each of two $d = 4$-dimensional posterior distributions $\mathbb{P}$: (1) a posterior over the parameters of the *Goodwin model* of oscillatory enzymatic control (Goodwin, 1965) and (2) a posterior over the parameters of the *Lotka-Volterra model* of oscillatory predator-prey evolution (Lotka, 1925; Volterra, 1926). For each of these targets, Riabiz et al. (2020b) provide $2 \times 10^6$ sample points from the following four MCMC algorithms: Gaussian random walk (RW), adaptive Gaussian random walk (adaRW, Haario et al., 1999), Metropolis-adjusted Langevin algorithm (MALA, Roberts & Tweedie, 1996), and pre-conditioned MALA (pMALA, Girolami & Calderhead, 2011).

**Hinch experiments**   Riabiz et al. (2020b) also provide the output of two independent Gaussian random walk MCMC chains targeting each of two $d = 38$-dimensional posterior distributions $\mathbb{P}$: (1) a posterior over the parameters of the Hinch model of calcium signalling in cardiac cells (Hinch et al., 2004) and (2) a tempered version of the same posterior, as defined by Riabiz et al. (2020a, App. S5.4).

**Burn-in and standard thinning**   We discard the initial burn-in points of each chain using the maximum burn-in period reported in Riabiz et al. (2020a, Tabs. S4 & S6, App. S5.4). Furthermore, we also normalize each Hinch chain by subtracting the post-burn-in sample mean and dividing each coordinate by its post-burn-in sample standard deviation. To obtain an input sequence $\mathcal{S}_{\text{in}}$ of length

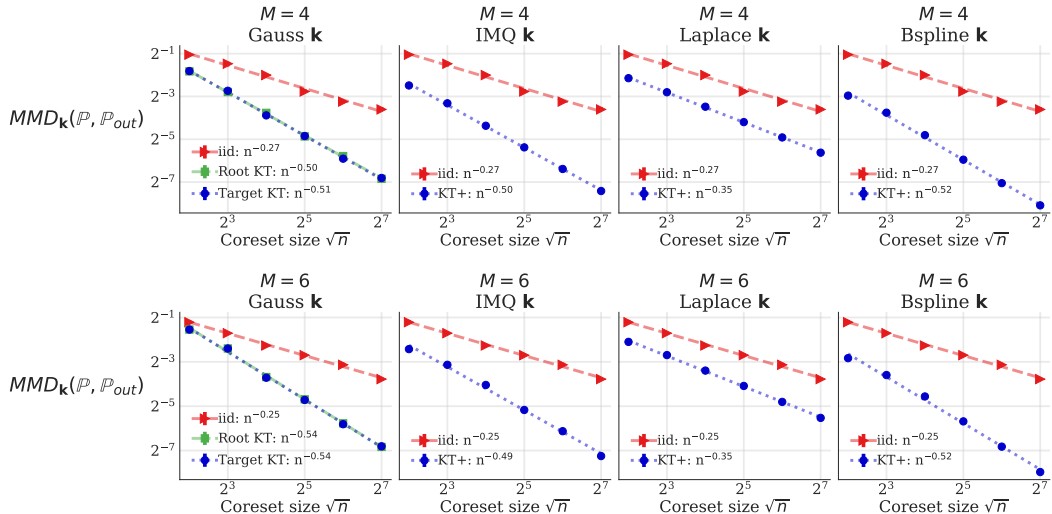

**Figure 5: Kernel thinning versus i.i.d. sampling.** For mixture of Gaussians $\mathbb{P}$ with $M \in \{4, 6\}$ components and the kernel choices of Sec. 4, the TARGET KT with GAUSS **k** provides comparable $\mathrm{MMD}_{\mathbf{k}}(\mathbb{P}, \mathbb{P}_{\mathrm{out}})$ error to the ROOT KT, and both provide an $n^{-\frac{1}{2}}$ decay rate improving significantly over the $n^{-\frac{1}{4}}$ decay rate from i.i.d. sampling. For the other kernels, KT+ provides a decay rate close to $n^{-\frac{1}{2}}$ for IMQ and B-SPLINE **k**, and $n^{-0.35}$ for LAPLACE **k**, providing an excellent agreement with the MMD guarantees provided by our theory. See Sec. 4 for further discussion.

$n$ to be fed into a thinning algorithm, we downsample the remaining even indices of points using standard thinning (odd indices are held out). When applying standard thinning to any Markov chain output, we adopt the convention of keeping the final sample point.

The selected burn-in periods for the Goodwin task were 820,000 for RW; 824,000 for adaRW; 1,615,000 for MALA; and 1,475,000 for pMALA. The respective numbers for the Lotka-Volterra task were 1,512,000 for RW; 1,797,000 for adaRW; 1,573,000 for MALA; and 1,251,000 for pMALA.

**Additional remarks on Fig. 3** When a Markov chain is fast mixing (as in the Goodwin and Lotka-Volterra examples), we expect standard thinning to have $\Omega(n^{-\frac{1}{4}})$ error. However, when the chain is slow mixing, standard thinning can enjoy a faster rate of decay due to a certain degeneracy of the chain that leads it to lie close to a one-dimensional curve. In the Hinch figures, we observe these better-than-i.i.d. rates of decay for standard thinning, but, remarkably, KT+ still offers improvements in both MMD and integration error. Moreover, in this setting, every additional point discarded via improved compression translates into thousands of CPU hours saved in downstream heart-model simulations.

## J  UPPER BOUNDS ON RKHS COVERING NUMBERS

In this section, we state several results on covering bounds for RKHSes for both generic and specific kernels. We then use these bounds with Thm. 2 (or Tab. 2) to establish MMD guarantees for the output of generalized kernel thinning as summarized in Tab. 3.

We first state covering number bounds for RKHS associated with generic kernels, that are either (a) analytic, or (b) finitely many times differentiable. These results follow essentially from Sun & Zhou (2008); Steinwart & Christmann (2008), but we provide a proof in App. J.2 for completeness.

**Proposition 2 (Covering numbers for analytic and differentiable kernels)** *The following results hold true.*

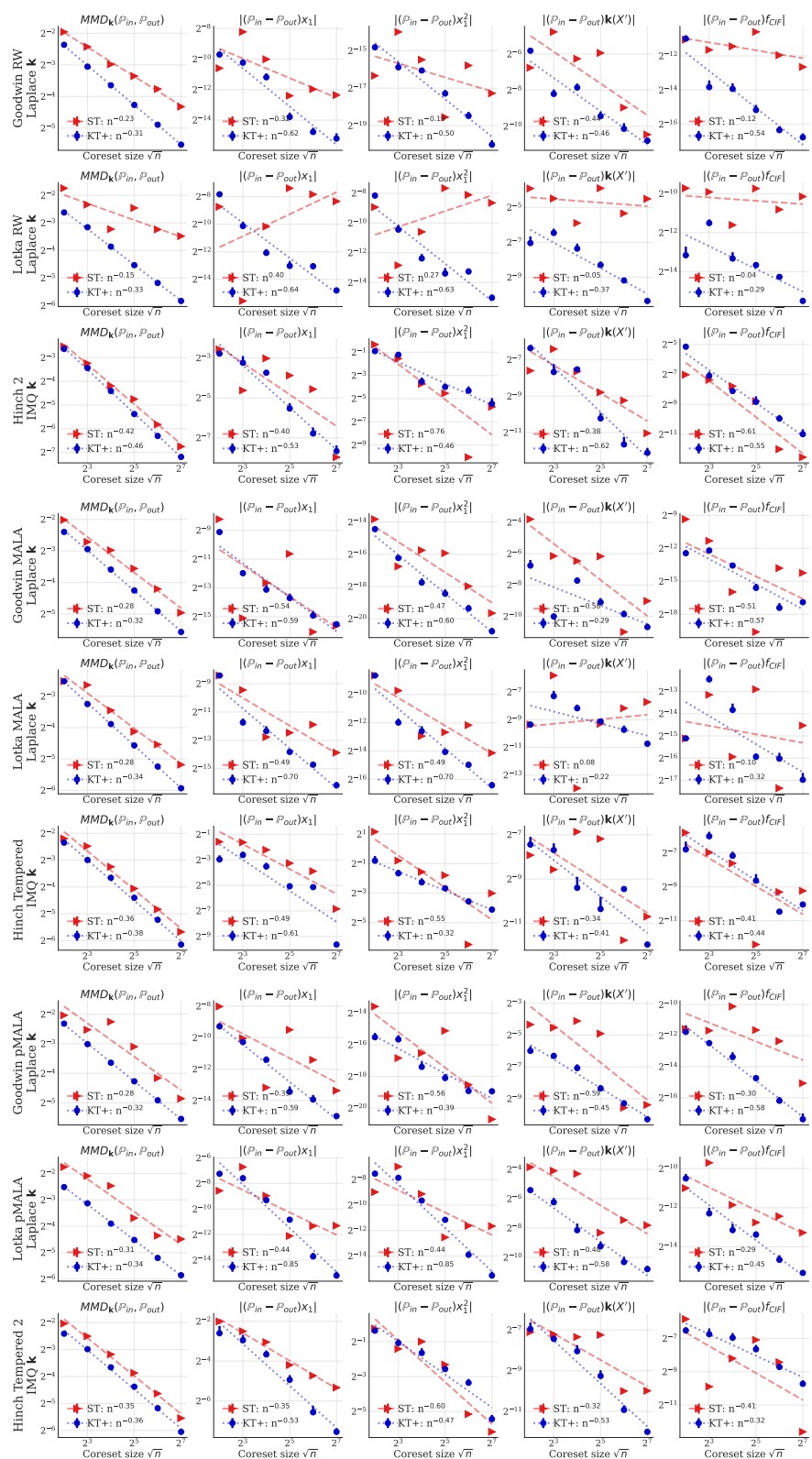

**Figure 6: Kernel thinning+ (KT+) vs. standard MCMC thinning (ST).** For kernels without fast-decaying square-roots, KT+ improves MMD and integration error decay rates in each posterior inference task.

(a) **Analytic kernels:** *Suppose that* $\mathbf{k}(x, y) = \kappa(\|x - y\|_2^2)$ *for* $\kappa : \mathbb{R}_+ \to \mathbb{R}$ *real-analytic with convergence radius* $R_\kappa$*, that is,*

$$\left| \tfrac{1}{j!} \kappa_+^{(j)}(0) \right| \le C_\kappa (2/R_\kappa)^j \quad \text{for all} \quad j \in \mathbb{N}_0 \tag{36}$$

*for some constant* $C_\kappa$*, where* $\kappa_+^{(j)}$ *denotes the right-sided* $j$*-th derivative of* $\kappa$*. Then for any set* $\mathcal{A} \subset \mathbb{R}^d$ *and any* $\varepsilon \in (0, \tfrac{1}{2})$*, we have*

$$\mathcal{M}_\mathbf{k}(\mathcal{A}, \varepsilon) \le \mathcal{N}_2(\mathcal{A}, r^\dagger/2) \cdot \left( 4 \log(1/\varepsilon) + 2 + 4 \log(16\sqrt{C_\kappa} + 1) \right)^{d+1}, \tag{37}$$

$$\text{where } r^\dagger \triangleq \min\left( \tfrac{\sqrt{R_\kappa}}{2d}, \sqrt{R_\kappa + D_\mathcal{A}^2} - D_\mathcal{A} \right), \text{ and } D_\mathcal{A} \triangleq \max_{x,y \in \mathcal{A}} \|x - y\|_2. \tag{38}$$

(b) **Differentiable kernels:** *Suppose that for* $\mathcal{X} \subset \mathbb{R}^d$*, the kernel* $\mathbf{k} : \mathcal{X} \times \mathcal{X} \to \mathbb{R}$ *is* $s$*-times continuously differentiable, i.e., all partial derivatives* $\partial^{\alpha,\alpha} \mathbf{k} : \mathcal{X} \times \mathcal{X} \to \mathbb{R}$ *exist and are continuous for all multi-indices* $\alpha \in \mathbb{N}_0^d$ *with* $|\alpha| \le s$*. Then, for any closed Euclidean ball* $\bar{\mathcal{B}}_2(r)$ *contained in* $\mathcal{X}$ *and any* $\varepsilon > 0$*, we have*

$$\mathcal{M}_\mathbf{k}(\bar{\mathcal{B}}_2(r), \varepsilon) \le c_{s,d,\mathbf{k}} \cdot r^d \cdot (1/\varepsilon)^{d/s}, \tag{39}$$

*for some constant* $c_{s,d,\mathbf{k}}$ *that depends only on on* $s, d$ *and* $\mathbf{k}$*.*

Next, we state several explicit bounds on covering numbers for several popular kernels. See App. J.3 for the proof.

**Proposition 3 (Covering numbers for specific kernels)** *The following statements hold true.*

(a) *When* $\mathbf{k} = \text{GAUSS}(\sigma)$*, we have*

$$\mathcal{M}_\mathbf{k}(\mathcal{B}_2(r), \varepsilon) \le C_{\text{Gauss},d} \cdot \left( \tfrac{\log(4/\varepsilon)}{\log \log(4/\varepsilon)} \right)^d \log(1/\varepsilon) \cdot \begin{cases} 1 & \text{when } r \le \tfrac{1}{\sqrt{2}\sigma}, \\ (3\sqrt{2}r\sigma)^d & \text{otherwise,} \end{cases} \tag{40}$$

$$\text{where } C_{\text{GAUSS},d} \triangleq \binom{4e+d}{d} e^{-d} \le \begin{cases} 4.3679 & \text{for } d = 1 \\ 0.05 \cdot d^{4e} e^{-d} & \text{for } d \ge 2 \end{cases} \le 30 \text{ for all } d \ge 1. \tag{41}$$

(b) *When* $\mathbf{k} = \text{MATÉRN}(\nu, \gamma)$*,* $\nu \ge \tfrac{d}{2} + 1$*, then for some constant* $C_{\text{MATÉRN},\nu,\gamma,d}$*, we have*

$$\mathcal{M}_\mathbf{k}(\mathcal{B}_2(r), \varepsilon) \le C_{\text{MATÉRN},\nu,\gamma,d} \cdot r^d \cdot (1/\varepsilon)^{d/\lfloor \nu - \frac{d}{2} \rfloor}. \tag{42}$$

(c) *When* $\mathbf{k} = \text{IMQ}(\nu, \gamma)$*, we have*

$$\mathcal{M}_\mathbf{k}(\mathcal{B}_2(r), \varepsilon) \le (1 + \tfrac{4r}{\tilde{r}})^d \cdot \left( 4 \log(1/\varepsilon) + 2 + C_{\text{IMQ},\nu,\gamma} \right)^{d+1}, \tag{43}$$

$$\text{where } C_{\text{IMQ},\nu,\gamma} \triangleq 4 \log\left( 16 \tfrac{(2\nu+1)^{\nu+1}}{\gamma^{2\nu}} + 1 \right), \text{ and } \tilde{r} \triangleq \min\left( \tfrac{\gamma}{2d}, \sqrt{\gamma^2 + 4r^2} - 2r \right). \tag{44}$$

(d) *When* $\mathbf{k} = \text{SINC}(\theta)$*, then for* $\varepsilon \in (0, \tfrac{1}{2})$*, we have*

$$\mathcal{M}_\mathbf{k}([-r, r]^d, \varepsilon) \le d \cdot (1 + \tfrac{4r}{\tilde{r}_\theta}) \cdot \left( 4 \log(d/\varepsilon) + 2 + 4 \log 17 \right)^2, \tag{45}$$

$$\text{where} \quad \tilde{r}_\theta \triangleq \min\left( \tfrac{\sqrt{3}}{|\theta|}, \sqrt{\tfrac{12}{\theta^2} + 4r^2} - 2r \right). \tag{46}$$

(e) *When* $\mathbf{k} = \text{B-SPLINE}(2\beta + 1, \gamma)$*, then for some universal constant* $C_{\text{B-SPLINE}}$*, we have*

$$\mathcal{M}_\mathbf{k}([-\tfrac{1}{2}, \tfrac{1}{2}]^d, \varepsilon) \le d \cdot \max(\gamma, 1) \cdot C_{\text{B-SPLINE}} \cdot (d/\varepsilon)^{\frac{1}{\beta + \frac{1}{2}}}. \tag{47}$$

## J.1 AUXILIARY RESULTS ABOUT RKHS AND EUCLIDEAN COVERING NUMBERS

In this section, we collect several results regarding the covering numbers of Euclidean and RKHS spaces that come in handy for our proofs. These results can also be of independent interest.

We start by defining the notion of restricted kernel and its unit ball (Rudi et al. (2020, Prop. 8)). For $\mathcal{X} \subset \mathbb{R}^d$, let $|_\mathcal{X}$ denotes the restriction operator. That is, for any function $f : \mathbb{R}^d \to \mathbb{R}$, we have $f|_\mathcal{X} : \mathcal{X} \to \mathbb{R}$ such that $f|_\mathcal{A}(x) = f(x)$ for $x \in \mathcal{X}$.

**Definition 3 (Restricted kernel and its RKHS)** *Consider a kernel $\mathbf{k}$ defined on $\mathbb{R}^d \times \mathbb{R}^d$ with the corresponding RKHS $\mathcal{H}$, any set $\mathcal{X} \subset \mathbb{R}^d$. The restricted kernel $\mathbf{k}|_{\mathcal{X}}$ is defined as*

$$\mathbf{k}|_{\mathcal{X}} : \mathcal{X} \times \mathcal{X} \to \mathbb{R} \quad \text{such that} \quad \mathbf{k}|_{\mathcal{X}}(x,y) \triangleq \mathbf{k}|_{\mathcal{X} \times \mathcal{X}}(x,y) = \mathbf{k}(x,y) \quad \text{for all} \quad x,y \in \mathcal{X},$$

*and $\mathcal{H}|_{\mathcal{X}}$ denotes its RKHS. For $f \in \mathcal{H}|_{\mathcal{X}}$, the restricted RKHS norm is defined as follows:*

$$\|f\|_{\mathbf{k}|_{\mathcal{X}}} = \inf_{h \in \mathcal{H}} \|h\|_{\mathbf{k}} \quad \text{such that} \quad h|_{\mathcal{X}} = f.$$

*Furthermore, we use $\mathcal{B}_{\mathbf{k}|_{\mathcal{X}}} \triangleq \{f \in \mathcal{H}|_{\mathcal{X}} : \|f\|_{\mathbf{k}|_{\mathcal{X}}} \le 1\}$ to denote the unit ball of the RKHS corresponding to this restricted kernel.*

In this notation, the unit ball of unrestricted kernel satisfies $\mathcal{B}_{\mathbf{k}} \triangleq \mathcal{B}_{\mathbf{k}|_{\mathbb{R}^d}}$. Now, recall the RKHS covering number definition from Def. 1. In the sequel, we also use the covering number of the restricted kernel defined as follows:

$$\mathcal{N}_{\mathbf{k}}^{\dagger}(\mathcal{X}, \varepsilon) = \mathcal{N}_{\mathbf{k}|_{\mathcal{X}}}(\mathcal{X}, \varepsilon), \tag{48}$$

that is $\mathcal{N}_{\mathbf{k}}^{\dagger}(\mathcal{X}, \varepsilon)$ denotes the minimum cardinality over all possible covers $\mathcal{C} \subset \mathcal{B}_{\mathbf{k}|_{\mathcal{X}}}$ that satisfy

$$\mathcal{B}_{\mathbf{k}|_{\mathcal{X}}} \subset \bigcup_{h \in \mathcal{C}} \left\{ g \in \mathcal{B}_{\mathbf{k}|_{\mathcal{X}}} : \sup_{x \in \mathcal{X}} |h(x) - g(x)| \le \varepsilon \right\}.$$

With this notation in place, we now state a result that relates the covering numbers $\mathcal{N}^{\dagger}$ (48) and $\mathcal{N}$ Def. 1.

**Lemma 2 (Relation between restricted and unrestricted RKHS covering numbers)** *We have*

$$\mathcal{N}_{\mathbf{k},\varepsilon}(\mathcal{X}) \le \mathcal{N}_{\mathbf{k},\varepsilon}^{\dagger}(\mathcal{X})$$

**Proof** Rudi et al. (2020, Prop. 8(d,f)) imply that there exists a bounded linear extension operator $E : \mathcal{H}|_{\mathcal{X}} \to \mathcal{H}$ with operator norm bounded by 1, which when combined with Steinwart & Christmann (2008, eqns. (A.38), (A.39)) yields the claim. □

Next, we state results that relate RKHS covering numbers for a change of domain for a shift-invariant kernel. We use $\mathcal{B}_{\|\cdot\|}(x; r) \triangleq \{y \in \mathbb{R}^d : \|x - y\| \le r\}$ to denote the $r$ radius ball in $\mathbb{R}^d$ defined by the metric induced by a norm $\|\cdot\|$.

**Definition 4 (Euclidean covering numbers)** *Given a set $\mathcal{X} \subset \mathbb{R}^d$, a norm $\|\cdot\|$, and a scalar $\varepsilon > 0$, we use $\mathcal{N}_{\|\cdot\|}(\mathcal{X}, \varepsilon)$ to denote the $\varepsilon$-covering number of $\mathcal{X}$ with respect to $\|\cdot\|$-norm. That is, $\mathcal{N}_{\|\cdot\|}(\mathcal{X}, \varepsilon)$ denotes the minimum cardinality over all possible covers $\mathcal{C} \subset \mathcal{X}$ that satisfy*

$$\mathcal{X} \subset \cup_{z \in \mathcal{C}} \mathcal{B}_{\|\cdot\|}(z; \varepsilon).$$

*When $\|\cdot\| = \|\cdot\|_q$ for some $q \in [1, \infty]$, we use the shorthand $\mathcal{N}_q \triangleq \mathcal{N}_{\|\cdot\|_q}$.*

**Lemma 3 (Relation between RKHS covering numbers on different domains)** *Given a shift-invariant kernel $\mathbf{k}$, a norm $\|\cdot\|$ on $\mathbb{R}^d$, and any set $\mathcal{X} \subset \mathbb{R}^d$, we have*

$$\mathcal{N}_{\mathbf{k}}^{\dagger}(\mathcal{X}, \varepsilon) \le \left[ \mathcal{N}_{\mathbf{k}}^{\dagger}(\mathcal{B}_{\|\cdot\|}, \varepsilon) \right]^{\mathcal{N}_{\|\cdot\|}(\mathcal{X}, 1)}.$$

**Proof** Let $\mathcal{C} \subset \mathcal{X}$ denote the cover of minimum cardinality such that

$$\mathcal{X} \subseteq \bigcup_{z \in \mathcal{C}} \mathcal{B}_{\|\cdot\|}(z, 1).$$

We then have

$$\mathcal{N}_{\mathbf{k}}^{\dagger}(\mathcal{X}, \varepsilon) \stackrel{(i)}{\le} \prod_{z \in \mathcal{C}} \mathcal{N}_{\mathbf{k}}^{\dagger}(\mathcal{B}_{\|\cdot\|}(z, 1), \varepsilon) \stackrel{(ii)}{\le} \prod_{z \in \mathcal{C}} \mathcal{N}_{\mathbf{k}}^{\dagger}(\mathcal{B}_{\|\cdot\|}, \varepsilon) \le \left[ \mathcal{N}_{\mathbf{k}}^{\dagger}(\mathcal{B}_{\|\cdot\|}, \varepsilon) \right]^{|\mathcal{C}|},$$

where step (i) follows by applying Steinwart & Fischer (2021, Lem. 3.11),[5] and step (ii) follows by applying Steinwart & Fischer (2021, Lem. 3.10). The claim follows by noting that $\mathcal{C}$ denotes a cover of minimum cardinality, and hence by definition $|\mathcal{C}| = \mathcal{N}_{\|\cdot\|}(\mathcal{X}, 1)$. □

---

[5] Steinwart & Fischer (2021, Lem. 3.11) is stated for disjoint partition of $\mathcal{X}$ in two sets, but the argument can be repeated for any finite cover of $\mathcal{X}$.

**Lemma 4 (Covering number for product kernel)** *Given* $\mathcal{X} \subset \mathbb{R}$ *and a reproducing kernel* $\kappa :$ *$\mathcal{X} \times \mathcal{X} \to \mathbb{R}$, consider the product kernel* $\mathbf{k} \triangleq \kappa^{\otimes d} : \mathcal{X}^{\otimes 2d} \to \mathbb{R}$ *defined as*

$$\mathbf{k}(x,y) = \prod_{i=1}^d \kappa(x_i, y_i) \quad for \quad x, y \in \mathcal{X}^{\otimes d} \triangleq \underbrace{\mathcal{X} \times \mathcal{X} \dots \times \mathcal{X}}_{d \text{ times}} \subset \mathbb{R}^d.$$

*Then the covering numbers of the two kernels are related as follows:*

$$\mathcal{N}_{\mathbf{k}}^\dagger(\mathcal{X}^{\otimes d}, \varepsilon) \le \left[ \mathcal{N}_\kappa^\dagger(\mathcal{X}, \varepsilon/(d\|\kappa\|_\infty^{\frac{d-1}{2}})) \right]^d. \tag{49}$$

**Proof**   Let $\mathcal{H}$ denote the RKHS corresponding to $\kappa$. Then the RKHS corresponding to the kernel $\mathbf{k}$ is given by the tensor product $\mathcal{H}_\mathbf{k} \triangleq \mathcal{H} \times \mathcal{H} \times \dots \times \mathcal{H}$ Berlinet & Thomas-Agnan (2011, Sec. 4.6), i.e., for any $f \in \mathcal{H}_\mathbf{k}$, there exists $f_1, f_2, \dots, f_d \in \mathcal{H}$ such that

$$f(x) = \prod_{i=1}^d f_i(x_i) \quad \text{for all} \quad x \in \mathcal{X}^{\otimes d}. \tag{50}$$

Let $\mathcal{C}_\kappa(\mathcal{X}, \varepsilon) \subset \mathcal{B}_\kappa$ denote an $\varepsilon$-cover of $\mathcal{B}_\kappa$ in $L^\infty$-norm (Def. 1). Then for each $f_i \in \mathcal{H}$, we have $\widetilde{f}_i \in \mathcal{C}_\kappa(\mathcal{X}, \varepsilon)$ such that

$$\sup_{z \in \mathcal{X}} \left| f_i(z) - \widetilde{f}_i(z) \right| \le \varepsilon. \tag{51}$$

Now, we claim that for every $f \in \mathcal{B}_\mathbf{k}$, there exists $g \in \mathcal{C}_\mathbf{k} \triangleq (\mathcal{C}_\kappa(\mathcal{X}, \varepsilon))^{\otimes d}$ such that

$$\sup_{x \in \mathcal{X}^{\otimes d}} |f(x) - g(x)| \le d\varepsilon \|\kappa\|_\infty^{\frac{d-1}{2}}, \tag{52}$$

which immediately implies the claimed bound (49) on the covering number. We now prove the claim (52). For any fixed $f \in \mathcal{H}_\mathbf{k}$, let $f_i, \widetilde{f}_i$ denote the functions satisfying (50) and (51) respectively. Then, we prove our claim (52) with $g = \prod_{i=1}^d \widetilde{f}_i \in \mathcal{C}_\mathbf{k}$. Using the convention $\prod_{k=1}^0 \widetilde{f}_k(x_k) = 1$, we find that

$$\begin{aligned} |f(x) - g(x)| &= \left| \prod_{i=1}^d f_i(x_i) - \prod_{i=1}^d \widetilde{f}_i(x_i) \right| \\ &\le \sum_{i=1}^d \left| f_i(x_i) - \widetilde{f}_i(x_i) \right| \left| \prod_{j=i+1}^d f_j(x_j) \prod_{k=1}^{i-1} \widetilde{f}_k(x_k) \right| \\ &\overset{(51)}{\le} d\varepsilon \cdot \sup_{h \in \mathcal{B}_\kappa} \|h\|_\infty^{d-1} \le d\varepsilon \|\kappa\|_\infty^{\frac{d-1}{2}}, \end{aligned}$$

where in the last step we have used the following argument:

$$\sup_{z \in \mathcal{X}} h(x) = \sup_{z \in \mathcal{X}} \langle h, \kappa(z, \cdot) \rangle_\kappa \le \|h\|_\kappa \sqrt{\kappa(z, z)} \le \sqrt{\|\kappa\|_\infty} \quad \text{for any} \quad h \in \mathcal{B}_\kappa.$$

The proof is now complete. $\qquad\square$

**Lemma 5 (Relation between Euclidean covering numbers)** *We have*

$$\mathcal{N}_\infty(\mathcal{B}_2(r), 1) \le \frac{1}{\sqrt{\pi d}} \cdot \left[ (1 + \frac{2r}{\sqrt{d}}) \sqrt{2\pi e} \right]^d \quad for \ all \quad d \ge 1.$$

**Proof**   We apply Wainwright (2019, Lem. 5.7) with $\mathcal{B} = \mathcal{B}_2(r)$ and $\mathcal{B}' = \mathcal{B}_\infty(1)$ to conclude that

$$\mathcal{N}_\infty(\mathcal{B}_2(r), 1) \le \frac{\text{Vol}(2\mathcal{B}_2(r) + \mathcal{B}_\infty(1))}{\text{Vol}(\mathcal{B}_\infty(1))} \le \text{Vol}(\mathcal{B}_2(2r + \sqrt{d})) \le \frac{\pi^{d/2}}{\Gamma(\frac{d}{2}+1)} \cdot (2r + \sqrt{d})^d,$$

where $\text{Vol}(\mathcal{X})$ denotes the $d$-dimensional Euclidean volume of $\mathcal{X} \subset \mathbb{R}^d$, and $\Gamma(a)$ denotes the Gamma function. Next, we apply the following bounds on the Gamma function from Batir (2017, Thm. 2.2):

$$\Gamma(b+1) \ge (b/e)^b \sqrt{2\pi b} \text{ for any } b \ge 1, \quad \text{and} \quad \Gamma(b+1) \le (b/e)^b \sqrt{e^2 b} \text{ for any } b \ge 1.1.$$

Thus, we have

$$\mathcal{N}_\infty(\mathcal{B}_2(r), 1) \le \frac{\pi^{d/2}}{\sqrt{2\pi d}(\frac{d}{2e})^{d/2}} \cdot (2r + \sqrt{d})^d \le \frac{1}{\sqrt{\pi d}} \cdot \left[ (1 + \frac{2r}{\sqrt{d}}) \sqrt{2e\pi} \right]^d,$$

as claimed, and we are done. $\qquad\square$

## J.2 Proof of Prop. 2: Covering numbers for analytic and differentiable kernels

First we apply Lem. 2 so that it remains to establish the stated bounds simply on $\log \mathcal{N}_{\mathbf{k}}^{\dagger}(\mathcal{X}, \varepsilon)$.

**Proof of bound (37) in part (a)** The bound (37) for the real-analytic kernel is a restatement of Sun & Zhou (2008, Thm. 2) in our notation (in particular, after making the following substitutions in their notation: $R \leftarrow 1, C_0 \leftarrow C_\kappa, r \leftarrow R_\kappa, \mathcal{X} \leftarrow \mathcal{A}, \widetilde{r} \leftarrow r^\dagger, \eta \leftarrow \varepsilon, D \leftarrow D_{\mathcal{A}}^2, n \leftarrow d$). $\square$

**Proof of bound (39) for part (b):** Under these assumptions, Steinwart & Christmann (2008, Thm. 6.26) states that the $i$-th dyadic entropy number Steinwart & Christmann (2008, Def. 6.20) of the identity inclusion mapping from $\mathcal{H}|_{\bar{\mathcal{B}}_2(r)}$ to $L_{\bar{\mathcal{B}}_2(r)}^\infty$ is bounded by $c'_{s,d,\mathbf{k}} \cdot r^s i^{-s/d}$ for some constant $c'_{s,d,\mathbf{k}}$ independent of $\varepsilon$ and $r$. Given this bound on the entropy number, and applying Steinwart & Christmann (2008, Lem. 6.21), we conclude that the log-covering number $\log \mathcal{N}_{\mathbf{k}}^{\dagger}(\bar{\mathcal{B}}_2(r), \varepsilon)$ is bounded by $\ln 4 \cdot (c'_{s,d,\mathbf{k}} r^s / \varepsilon)^{d/s} = c_{s,d,\mathbf{k}} r^d \cdot (1/\varepsilon)^{d/s}$ as claimed. $\square$

## J.3 Proof of Prop. 3: Covering numbers for specific kernels

First we apply Lem. 2 so that it remains to establish the stated bounds in each part on the corresponding $\log \mathcal{N}_{\mathbf{k}}$.

**Proof for Gauss kernel: Part (a)** The bound (40) for the Gaussian kernel follows directly from Steinwart & Fischer (2021, Eqn. 11) along with the discussion stated just before it. Furthermore, the bound (41) for $C_{\text{Gauss},d}$ are established in Steinwart & Fischer (2021, Eqn. 6), and in the discussion around it. $\square$

**Proof for Matérn kernel: Part (b)** We claim that $\text{MATÉRN}(\nu, \gamma)$ is $\lfloor \nu - \frac{d}{2} \rfloor$-times continuously differentiable which immediately implies the bound (42) using Prop. 2(b).

To prove the differentiability, we use Fourier transform of Matérn kernels. For $\mathbf{k} = \text{MATÉRN}(\nu, \gamma)$, let $\kappa : \mathbb{R}^d \to \mathbb{R}$ denote the function such that noting that $\mathbf{k}(x, y) = \kappa(x - y)$. Then using the Fourier transform of $\kappa$ from Wendland (2004, Thm 8.15), and noting that $\kappa$ is real-valued, we can write

$$\mathbf{k}(x, y) = c_{\mathbf{k},d} \int \cos(\omega^\top (x - y))(\gamma^2 + \|\omega\|_2^2)^{-\nu} d\omega$$

for some constant $c_{\mathbf{k},d}$ depending only on the kernel parameter, and $d$ (due to the normalization of the kernel, and the Fourier transform convention). Next, for any multi-index $a \in \mathbb{N}_0^d$, we have

$$\left| \partial^{a,a} \cos(\omega^\top (x - y))(\gamma^2 + \|\omega\|_2^2)^{-\nu} \right| \le \prod_{j=1}^d \omega_j^{2a_j} (\gamma^2 + \|\omega\|_2^2)^{-\nu} \le \frac{\|\omega\|_2^{2\sum_{j=1}^d a_j}}{(\gamma^2 + \|\omega\|_2^2)^\nu},$$

where $\partial^{a,a}$ denotes the partial derivative of order $a$. Moreover, we have

$$\int \frac{\|\omega\|_2^{2\sum_{j=1}^d a_j}}{(\gamma^2 + \|\omega\|_2^2)^\nu} d\omega = c_d \int_{r>0} r^{d-1} \frac{r^{2\sum_{j=1}^d a_j}}{(\gamma^2 + r^2)^\nu} dr \le c_d \int_{r>0} r^{d-1+2\sum_{j=1}^d a_j - 2\nu} \overset{(i)}{<} \infty,$$

where step (i) holds whenever

$$d - 1 + 2\sum_{j=1}^d a_j - 2\nu < -1 \iff \sum_{j=1}^d a_j < \nu - \frac{d}{2}.$$

Then applying Newey & McFadden (1994, Lemma 3.6), we conclude that for all multi-indices $a$ such that $\sum_{j=1}^d a_j \le \lfloor \nu - \frac{d}{2} \rfloor$, the partial derivative $\partial^{a,a}\mathbf{k}$ exists and is given by

$$c_{\mathbf{k},d} \int \partial^{a,a} \cos(\omega^\top (x - y))(\gamma^2 + \|\omega\|_2^2)^{-\nu} d\omega,$$

and we are done. $\square$

**Proof for IMQ kernel: Part (c)** The bounds (43) and (44) follow from Sun & Zhou (2008, Ex. 3), and noting that $\mathcal{N}_2(\mathcal{B}_2(r), \widetilde{r}/2)$ is bounded by $(1 + \frac{4r}{\widetilde{r}})^d$ (Wainwright, 2019, Lem. 5.7). $\square$

**Proof for SINC kernel: Part (d)** For $\mathbf{k} = \text{SINC}(\theta)$, we can write $\mathbf{k}(x, y) = \prod_{i=1}^d \kappa_\theta(x_i - y_i)$ for $\kappa_\theta : \mathbb{R} \to \mathbb{R}$ defined as $\kappa_\theta(t) = \frac{\sin(\theta t)}{\theta t} \stackrel{(i)}{=} \frac{\sin(|\theta t|)}{|\theta t|}$, where step (i) follows from the fact that $t \mapsto \sin t / t$ is an even function. Thus, we can apply Lem. 4. Given the bound (49), and noting that $\|\kappa_\theta\|_\infty = 1$, it suffices to establish the univariate version of the bound (45), namely,

$$\mathcal{M}_{\mathbf{k}}([-r, r], \varepsilon) \leq (1 + \tfrac{4r}{r_\theta}) \cdot \left(4 \log(1/\varepsilon) + 2 + 4 \log 17\right)^2.$$

To do so, we claim that univariate SINC kernel is an analytic kernel that satisfies the condition (36) of Prop. 2(a) with $\kappa(t) = \text{SINC}(\theta \sqrt{t})$, $R_\kappa = \frac{12}{\theta^2}$, and $C_\kappa = 1$; and thus applying the bounds (37) and (38) from Prop. 2(a) with $\mathcal{A} = \mathcal{B}_2^d(r)$ yields the claimed bound (45) and (46). To verify the condition (36) with the stated parameters, we note that

$$\kappa(t) = \text{SINC}(\theta \sqrt{t}) = \frac{1}{|\theta|\sqrt{t}} \sum_{j=0}^\infty \frac{1}{(2j+1)!} \cdot (\theta\sqrt{t})^{2j+1} = \sum_{j=0}^\infty \frac{1}{(2j+1)!} \cdot (\theta\sqrt{t})^{2j}$$
$$= \sum_{j=0}^\infty \frac{1}{(2j+1)!} \cdot \theta^{2j} \cdot t^j$$

which implies

$$\left|\kappa_+^{(j)}(0)\right| = \frac{1}{(2j+1)!} \cdot \theta^{2j} j! \leq (2/R_\kappa)^j j! \quad \text{for} \quad R_\kappa \triangleq \frac{2}{\theta^2} \cdot \inf_{j \geq 1}((2j+1)!)^{1/j} = \frac{12}{\theta^2},$$

and we are done. $\square$

**Proof for B-SPLINE kernel: Part (e)** For $\mathbf{k} = \text{B-SPLINE}(2\beta + 1, \gamma)$, we can write $\mathbf{k}(x, y) = \prod_{i=1}^d \kappa_{\beta,\gamma}((x_i - y_i))$ for $\kappa_\beta : \mathbb{R} \to \mathbb{R}$ defined as $\kappa_{\beta,\gamma}(t) = \mathfrak{B}_{2\beta+2}^{-1} \circledast^{2\beta+2} \mathbf{1}_{[-\frac{1}{2}, \frac{1}{2}]}(\gamma \cdot t)$, and thus we can apply Lem. 4. Given the bound (49), and noting that $\|\kappa_{\beta,\gamma}\|_\infty \leq 1$ (Dwivedi & Mackey (2021, Eqn. 107)), it suffices to establish the univariate version of the bound (47). Abusing notation and using $\kappa_{\beta,\gamma}$ to denote the univariate B-SPLINE$(2\beta + 1, \gamma)$ kernel, we find that

$$\log \mathcal{N}_{\kappa_{\beta,\gamma}}^\dagger([-\tfrac{1}{2}, \tfrac{1}{2}], \varepsilon) \stackrel{(i)}{\leq} \mathcal{N}_1([0, \gamma], 1) \cdot \log \mathcal{N}_{\kappa_{\beta,1}}^\dagger([-\tfrac{1}{2}, \tfrac{1}{2}], \varepsilon)$$
$$\stackrel{(ii)}{\leq} \max(\gamma, 1) \cdot C_{\text{B-SPLINE}} \cdot (1/\varepsilon)^{\frac{1}{\beta + \frac{1}{2}}},$$

where step (i) follows from Steinwart & Fischer (2021, Thm. 2.4, Sec. 3.3), and for step (ii) we use the fact that the unit-covering number of $[0, \gamma]$ is bounded by $\max(\gamma, 1)$, and apply the covering number bound for the univariate B-SPLINE kernel from Zhou (2003, Ex. 4) (by substituting $m = 2\beta + 2$ in their notation) along with the fact that $\log \mathcal{N}_{\kappa_{\beta,1}}^\dagger([-\tfrac{1}{2}, \tfrac{1}{2}], \varepsilon) = \log \mathcal{N}_{\kappa_{\beta,1}}^\dagger([0, 1], \varepsilon)$ since $\kappa_\beta$ is shift-invariant. $\square$

## K  PROOF OF TAB. 3 RESULTS

In Tab. 3, the stated results for all the entries in the TARGET KT column follow directly by substituting the covering number bounds from Prop. 3 in the corresponding entry along with the stated radii growth conditions for the target $\mathbb{P}$. (We substitute $m = \frac{1}{2} \log_2 n$ since we thin to $\sqrt{n}$ output size.) For the KT+ column, the stated result follows by either taking the minimum of the first two columns (whenever the ROOT KT guarantee applies) or using the POWER KT guarantee. First we remark how to always ensure a rate of at least $\mathcal{O}(n^{-\frac{1}{4}})$ even when the guarantee from our theorems are larger, using a suitable baseline procedure and then proceed with our proofs.

**Remark 2 (Improvement over baseline thinning)** *First we note that the* KT-SWAP *step ensures that, deterministically,* $\text{MMD}_{\mathbf{k}}(\mathcal{S}_{\text{in}}, \mathcal{S}_{\text{KT}}) \leq \text{MMD}_{\mathbf{k}}(\mathcal{S}_{\text{in}}, \mathcal{S}_{\text{base}})$ *and* $\text{MMD}_{\mathbf{k}}(\mathbb{P}, \mathcal{S}_{\text{KT}}) \leq 2\,\text{MMD}_{\mathbf{k}}(\mathbb{P}, \mathcal{S}_{\text{in}}) + \text{MMD}_{\mathbf{k}}(\mathbb{P}, \mathcal{S}_{\text{base}})$ *for* $\mathcal{S}_{\text{base}}$ *a baseline thinned coreset of size* $\frac{n}{2^m}$ *and any target* $\mathbb{P}$. *For example if the input and baseline coresets are drawn i.i.d. and* $\mathbf{k}$ *is bounded, then*

$\mathrm{MMD}_{\mathbf{k}}(\mathcal{S}_{\mathrm{in}}, \mathcal{S}_{\mathrm{KT}})$ *and* $\mathrm{MMD}_{\mathbf{k}}(\mathbb{P}, \mathcal{S}_{\mathrm{KT}})$ *are* $\mathcal{O}(\sqrt{2^m/n})$ *with high probability (Tolstikhin et al., 2017, Thm. A.1), even if the guarantee of Thm. 2 is larger. As a consequence, in all well-defined KT variants, we can guarantee a rate of* $n^{-\frac{1}{4}}$ *for* $\mathrm{MMD}_{\mathbf{k}}(\mathcal{S}_{\mathrm{in}}, \mathcal{S}_{\mathrm{KT}})$ *when the output size is* $\sqrt{n}$ *simply by using baseline as i.i.d. thinning in the* KT-SWAP *step.*

**GAUSS kernel** The TARGET KT guarantee follows by substituting the covering number bound for the Gaussian kernel from Prop. 3(a) in (6), and the ROOT KT guarantee follows directly from Dwivedi & Mackey (2021, Tab. 2). Putting the guarantees for the ROOT KT and TARGET KT together (and taking the minimum of the two) yields the guarantee for KT+.

**IMQ kernel** The TARGET KT guarantee follows by putting together the covering bound Prop. 3(c) and the MMD bounds (6).

For the ROOT KT guarantee, we use a square-root dominating kernel $\widetilde{\mathbf{k}}_{\mathrm{rt}}$ IMQ$(\nu', \gamma')$ Dwivedi & Mackey (2021, Def.2) as suggested by Dwivedi & Mackey (2021). Dwivedi & Mackey (2021, Eqn.(117)) shows that $\widetilde{\mathbf{k}}_{\mathrm{rt}}$ is always defined for appropriate choices of $\nu', \gamma'$. The best ROOT KT guarantees are obtained by choosing largest possible $\nu'$ (to allow the most rapid decay of tails), and Dwivedi & Mackey (2021, Eqn.(117)) implies with $\nu < \frac{d}{2}$, the best possible parameter satisfies $\nu' \leq \frac{d}{4} + \frac{\nu}{2}$. For this parameter, some algebra shows that $\max(\mathfrak{R}^{\dagger}_{\widetilde{\mathbf{k}}_{\mathrm{rt}}, n} \mathfrak{R}_{\widetilde{\mathbf{k}}_{\mathrm{rt}}, n}) \precsim_{d, \nu, \gamma} n^{1/2\nu}$, leading to a guarantee worse than $n^{-\frac{1}{4}}$, so that the guarantee degenerates to $n^{-\frac{1}{4}}$ using Rem. 2 for ROOT KT. When $\nu \geq \frac{d}{2}$, we can use a MATÉRN kernel as a square-root dominating kernel from Dwivedi & Mackey (2021, Prop. 3), and then applying the bounds for the kernel radii (17), and the inflation factor (19) for a generic Matérn kernel from Dwivedi & Mackey (2021, Tab. 3) leads to the entry for the ROOT KT stated in Tab. 2. The guarantee for KT+ follows by taking the minimum of the two.

**MATÉRN kernel** For TARGET KT, substituting the covering number bound from Prop. 3(b) in (6) with $R = \log n$ yields the MMD bound of order

$$\sqrt{\frac{\log n \cdot (\log n)^d \cdot n^{2\lfloor \nu - \frac{d}{2} \rfloor}}{n}}, \tag{53}$$

which is better than $n^{-\frac{1}{4}}$ only when $\nu > 3d/2$, and simplified to the entry in the Tab. 3 when we assume $\nu - \frac{d}{2}$ is an integer. When $\nu \leq 3d/2$, we can simply use baseline as i.i.d. thinning to obtain an order $n^{-\frac{1}{4}}$ MMD error as in Rem. 2.

The ROOT KT (and thereby KT+) guarantees for $\nu > d$ follow from Dwivedi & Mackey (2021, Tab. 2).

When $\nu \in (\frac{d}{2}, d]$, we use POWER KT with a suitable $\alpha$ to establish the KT+ guarantee. For MATÉRN$(\nu, \gamma)$ kernel, the $\alpha$-power kernel is given by MATÉRN$(\alpha\nu, \gamma)$ if $\alpha\nu > \frac{d}{2}$ (a proof of this follows from Def. 2 and Dwivedi & Mackey (2021, Eqns 71-72))). Since LAPLACE$(\sigma) =$ MATÉRN$(\frac{d+1}{2}, \sigma^{-1})$, we conclude that its $\alpha$-power kernel is defined for $\alpha > \frac{d}{d+1}$. And using the various tail radii (17), and the inflation factor (19) for a generic Matérn kernel from Dwivedi & Mackey (2021, Tab. 3), we conclude that $\widetilde{\mathfrak{M}}_{\alpha} \precsim_{d, \mathbf{k}_{\alpha}, \delta} \sqrt{\log n \log \log n}$, and $\max(\mathfrak{R}^{\dagger}_{\mathbf{k}_{\alpha}, n} \mathfrak{R}_{\mathbf{k}_{\alpha}, n}) \precsim_{d, \mathbf{k}_{\alpha}} \log n$, so that $\mathfrak{R}_{\max} = \mathcal{O}_{d, \mathbf{k}_{\alpha}}(\log n)$ (18) for SUBEXP $\mathbb{P}$ setting. Thus for this case, the MMD guarantee for $\sqrt{n}$ thinning with POWER KT (tracking only scaling with $n$) is

$$\left(\frac{2^m}{n} \|\mathbf{k}_{\alpha}\|_{\infty}\right)^{\frac{1}{2\alpha}} (2 \cdot \widetilde{\mathfrak{M}}_{\alpha})^{1 - \frac{1}{2\alpha}} \left(2 + \sqrt{\frac{(4\pi)^{d/2}}{\Gamma(\frac{d}{2}+1)} \cdot \mathfrak{R}^{\frac{d}{2}}_{\max} \cdot \widetilde{\mathfrak{M}}_{\alpha}}\right)^{\frac{1}{\alpha} - 1}$$

$$\precsim_{d, \mathbf{k}_{\alpha}, \delta} \left(\frac{1}{\sqrt{n}}\right)^{\frac{1}{2\alpha}} (\sqrt{c_n \log n})^{1 - \frac{1}{2\alpha}} \cdot ((\log n)^{\frac{d}{2} + \frac{1}{2}} \sqrt{c_n})^{\frac{1}{\alpha} - 1} = \left(\frac{c_n (\log n)^{1 + 2d(1-\alpha)}}{n}\right)^{\frac{1}{4\alpha}}$$

where $c_n = \log \log n$; and we thus obtain the corresponding entry (for KT+) stated in Tab. 3.

**SINC kernel** The guarantee for TARGET KT follows directly from substituting the covering number bounds from Prop. 3(d) in (6).

For the ROOT KT guarantee, we note that the square-root kernel construction of Dwivedi & Mackey (2021, Prop.2) implies that $\text{SINC}(\theta)$ itself is a square-root of $\text{SINC}(\theta)$ since the Fourier transform of SINC is a rectangle function on a bounded domain. However, the tail of the SINC kernel does not decay fast enough for the guarantee of Dwivedi & Mackey (2021, Thm. 1) to improve beyond the $n^{-\frac{1}{4}}$ bound of Dwivedi & Mackey (2021, Rem. 2) obtained when running ROOT KT with i.i.d. baseline thinning.

In this case, TARGET KT and KT+ are identical since $\mathbf{k}_{\text{rt}} = \mathbf{k}$.

**B-SPLINE kernel**     The guarantee for TARGET KT follows directly from substituting the covering number bounds from Prop. 3(d) in (6).

For B-SPLINE$(2\beta + 1, \gamma)$ kernel, using arguments similar to that in Dwivedi & Mackey (2021, Tab.4), we conclude that (up to a constant scaling) the $\alpha$-power kernel is defined to be B-SPLINE$(A + 1, \gamma)$ whenever $A \triangleq 2\alpha\beta + 2\alpha - 2 \in 2\mathbb{N}_0$. For odd $\beta$ we can always take $\alpha = \frac{1}{2}$ and B-SPLINE$(\beta + 1, \gamma)$ is a valid (up to a constant scaling) square-root kernel (Dwivedi & Mackey, 2021). For even $\beta$, we have to choose $\alpha \triangleq \frac{p+1}{\beta+1} \in (\frac{1}{2}, 1)$ by taking $p \in \mathbb{N}$ suitably, and the smallest suitable choice is $p = \lceil \frac{\beta-1}{2} \rceil = \frac{\beta}{2} \in \mathbb{N}$, which is feasible as long as $\beta \geq 2$. And, thus B-SPLINE$(\beta + 1, \gamma)$ is a suitable $\mathbf{k}_\alpha$ for B-SPLINE$(2\beta + 1)$ for even $\beta \geq 2$ with $\alpha = \frac{\beta+2}{2\beta+2} \in (\frac{1}{2}, 1)$. Whenever the $\alpha$-power kernel is defined, we can then apply the various tail radii (17), and the inflation factor (19) for the power B-SPLINE kernel from Dwivedi & Mackey (2021, Tab. 3) to obtain the MMD rates for POWER KT from Dwivedi & Mackey (2021, Tab. 2) (which remains the same as ROOT KT upto factors depending on $\alpha$ and $\beta$).

The guarantee for KT+ follows by taking the minimum MMD error for TARGET KT and ROOT KT for even $\beta$, and $\alpha$-POWER KT for odd $\beta$.

