# OpenReview forum: "Generalized Kernel Thinning"
_ICLR.cc/2022/Conference — ICLR 2022 Poster_

### Official Review · Reviewer_MW7b · 2021-11-02

**Correctness:** 4
**Technical Novelty And Significance:** 3
**Empirical Novelty And Significance:** 2
**Recommendation:** 6
**Confidence:** 2

**Main Review:**

The paper seems as an extension of results of a previous paper. The algorithm for finding a subsample is the same, the first part of it (KT-split) is applied with a general kernel (split kernel), which may differ from square root kernel. Therefore, an interesting question is how the analysis given in arXiv:2105.05842 corresponds with an analysis given in terms of RKHS covering number. Is a new analysis leads to the same thinning error, when applied to square root kernel (or, the square root is somehow special?).

Figure 2 shows that Root KT and Target KT give the same error as a function of coreset size. Is there any explanation of such behavior?

Experimental part convinces that root/target KT gives smaller errors than iid, though Fig 3 shows more complicated picture when KT+ is compared with ST (Hitch IMQ k). Is there any explanation of that?

**Summary Of The Paper:**

The paper's claims seem correct and sound. Claims are substantiated by a theoretical analysis, though the bulk of the paper's ideology comes from the previous publication (which the current paper extends). Experiments seem very similar to the previous paper's experiments.

**Summary Of The Review:**

I believe this is a borderline paper. Some aspects of it were published before. A new analysis is presented (based on RKHS covering number).

---

> ### Author Response · Authors · 2021-11-21
> **Author response**
>
> Thank you for the time you’ve spent reviewing our work and for your thoughtful feedback.  We are glad that you found all of our claims and statements well-supported and correct.  In case there is any confusion (since we are generalizing an existing algorithm and providing new analyses), we begin by clarifying that all of the material in our submission is new and that none has been previously published.  We now address each of your questions and concerns in turn.

---

> > ### Author Response · Authors · 2021-11-21
> > **Ques: Experimental part convinces that root/target KT gives smaller errors than iid, though Fig 3 shows more complicated picture when KT+ is compared with ST (Hitch IMQ k). Is there any explanation of that?**
> >
> > We will clarify that when a Markov chain is fast mixing (as in the Goodwin and Lotka examples), standard thinning has $\Omega(n^{-1/4})$ error, but when the chain is slow mixing it can have a faster rate of decay due to a certain degeneracy of the chain that leads it to lie close to a one-dimensional curve. In the Hinch example, we observe these better-than-iid rates of decay for standard thinning, but, remarkably KT+ still offers improvements in MMD and integration error (and in this setting, every additional point discarded via improved compression translates into 1000s of CPU hours saved in downstream heart-model simulation).

---

> > ### Author Response · Authors · 2021-11-21
> > **Ques: Figure 2 shows that Root KT and Target KT give the same error as a function of coreset size. Is there any explanation of such behavior?**
> >
> > Yes!  The guarantees we derive for Gaussian target KT in Table 3 nearly match those of D&M (2021) for Gaussian root KT, so we expected to see similar performance even though our target KT coresets make no explicit use of a square-root kernel.  We discuss this in the Generalized KT coresets paragraph on page 8.

---

> > ### Author Response · Authors · 2021-11-21
> > **How our analyses and results differ from those of D&M (2021)**
> >
> > D&M (2021) bound the MMD\_k of root KT by, first, bounding $\mathbb{MMD}\_{\mathbf{k}}$ in terms of square-root kernel $L^{\infty}$ error ($||(\mathbb{P}\_{{in}} - \mathbb{P}\_{{out}}) \mathbf{k}\_{{rt}}\Vert\_{\infty}$) and tail decay properties of $\mathbb P$ and $\mathbf{k}\_{{rt}}$ and, second, bounding the square-root kernel Linf error using high-probability tail bounds for the point evaluations $(\mathbb{P}\_{{in}} - \mathbb{P}\_{{out}}) \mathbf{k}\_{{rt}}(x)$ and a chaining argument.
> >
> > Our work introduces four different analyses, each of which is distinct from the analysis of D&M (2021):
> >
> > 1. Our Theorem 1 analysis bounds the integration error $\vert \mathbb{P}\_{{in}} f - \mathbb{P}\_{{out}} f \vert $ of each function f in the RKHS of $\mathbf{k}\_{{split}}$ directly, instead of targeting MMD. By modifying the original D&M algorithm to use the target kernel k instead of a square-root kernel, we obtain the first dimension-free guarantees for KT-SPLIT: integration error $\leq \sqrt{\log n} / \sqrt{n}$ with high probability.  An important implication is that target KT-SPLIT improves upon iid integration error even for small or moderate sample sizes and even in arbitrarily high dimensions.  In contrast, the guarantees of D&M (2021) all exhibit an exponential dependence on dimension as summarized in this table which bounds single function integration error of KT-SPLIT up to constants independent of d:
> >
> > ||D&M (2021) guarantee|Our Thm. 1 guarantee|iid sampling|
> > |-|-|-||
> > | Compact P |$c^d \sqrt{\log(n)/n}$|$\sqrt{\log(n)/n}$|$n^{-1/4}$|
> > | Sub-Gaussian P|$c^d\log^{(d/2+1)/2}(n)/\sqrt{n}$|$\sqrt{\log(n)/n}$|$n^{-1/4}$|
> > | Subexponential P|$c^d\log^{(d+1)/2}(n)/\sqrt{n}$|$\sqrt{\log(n)/n}$|$n^{-1/4}$|
> > | Heavy-tailed P|$c^d\log n/\sqrt{n^{1-d/\rho}}$|$\sqrt{\log(n)/n}$|$n^{-1/4}$|
> >
> >
> > As a result, in the best case, their guarantees only offer an improvement over iid sampling when the sample size n is exponentially large in the dimension (i.e., when $n \geq C^d$ for some $C > 1$), and, in the worst case (heavy tails with $\rho \leq 2d$ moments), their guarantees offer no improvement over iid sampling.
> >
> > Moreover, our use of the target kernel in place of D&M’s square-root kernel was essential for obtaining our dimension-free guarantees.  If we instead apply our single-function analysis to KT-SPLIT with the square-root kernel, we show in App. G that the corresponding integration error still grows exponentially in the dimension for the Gaussian and Matern kernels studied by D&M.
> >
> > 2. Our Theorem 2 target KT MMD guarantees are derived in a completely different and more direct way than those of D&M (2021): we use a covering number argument for the unit ball of the RKHS of k combined with the individual function guarantees of Theorem 1.  Our new bounds complement those of D&M and establish better-than-iid guarantees for an entire family of kernels not covered by the D&M analysis (analytic kernels without fast-decaying square-roots like sinc, slowly-decaying IMQ, and odd b-spline).
> >
> > 3. To bound power KT MMD in Theorem 3, we developed a novel power kernel analysis that interpolates between the root KT guarantees of D&M (2021) and the target KT guarantees of our Thm. 2 (recovering each when $\alpha = 1/2$ or $1$ respectively). This leads to the first better-than-iid MMD guarantees for the non-smooth kernels ruled out by the D&M algorithm (including Laplace and non-smooth Matern).
> >
> > 4. Our new KT+ analysis underlying Theorem 4 shows that KT-SPLIT applied to the sum of two kernels simultaneously yields the complementary guarantees of each kernel separately.

---

### Official Review · Reviewer_LtTV · 2021-11-02

**Correctness:** 4
**Technical Novelty And Significance:** 2
**Empirical Novelty And Significance:** 2
**Recommendation:** 8
**Confidence:** 2

**Main Review:**

I must apologise in advance that I am not an expert in the field of thinning, and the vast majority of the references were unfamiliar to me. However, I saw much more positives than negatives while reading the paper.

Firstly, the paper was written in a clear manner, with a very convincing motivation of the problem. The contributions were well outlined in the context of existing works. The significance of the theoretical results were clear, and as far as I could see, the results were correct. One aspect I particularly appreciated was the amount of effort put into analysing all the different types of distributions and kernels, and summarising them in various tables.

**Summary Of The Paper:**

The paper proposes "generalised kernel thinning", which builds on the "kernel thinning" (KT) algorithm of [Dwivedi and Mackey, 2021]. The main generalisation is to replace the square-root kernel in the algorithm with k_{split}, which results in applicability without the knowledge of the square-root kernel, and theoretical guarantees are proved. Moreover, two instantiations of this generalisation are proposed, the alpha-power KT and KT+.

**Summary Of The Review:**

The paper motivates the problem in a clear and convincing way even for a non-expert, and clearly outlines its contributions. The results are significant, and as far as I can see, correct and sound. I recommend this paper to be accepted.

---

> ### Author Response · Authors · 2021-11-21
> **Author response**
>
> Thank you for the time you’ve spent reviewing our work and for your thoughtful feedback.
> We are delighted that you found our results to be clearly significant and well-outlined and our presentation and motivation clear and convincing even for non-experts.

---

### Official Review · Reviewer_8dTF · 2021-11-03

**Correctness:** 4
**Technical Novelty And Significance:** 3
**Empirical Novelty And Significance:** 2
**Recommendation:** 5
**Confidence:** 3

**Main Review:**

It is hard to follow the ‘flow’ of this work. For instance, the definition of  KT-Split is only given in the appendix. It is preferable to recall briefly the principle and the intuition behind this algorithm in the main paper. The reader may ask some legitimate questions: what is the role of the kernel k_{split} in the algorithm? What are the benefits we gain from studying power kernel thinning?

The presentation of the paper is non-conventional. For example, Theorem 1 is followed by many remarks. The content of these remarks may fit in one or two paragraphs and the presentation would be more elegant.

**Summary Of The Paper:**

This article studies a generalization of kernel thinning previously proposed in Dwivedi and Mackey (2021). The authors proposed a new variant of kernel thinning that does not require the knowledge of the square root of the kernel, for which they proved single function guarantees as well as the MMD guarantees. Several numerical simulations were conducted to support the theoretical claims.

**Summary Of The Review:**

Up to my knowledge, the proposed algorithms and the corresponding theoretical guarantees are new in the community. However, the manuscript is heavily technical and dense. I am afraid that only readers familiar with kernel thinning may grasp the contribution of this paper.

---

> ### Author Response · Authors · 2021-11-21
> **Author response**
>
> Thank you for the time you’ve spent reviewing our work, for your thoughtful feedback, and for your appreciation of the novelty of our work.  We address each of your questions and concerns below.

---

> > ### Author Response · Authors · 2021-11-21
> > **The usefulness of power KT**
> >
> > Power KT
> >
> > 1. Ensures better-than-iid MMD for non-smooth kernels like the Laplace and non-smooth Matern kernels and
> > 2. Serves as a building block for KT+ which simultaneously enjoys the MMD improvements of power KT and the single-function integration error improvements of target KT.
> >
> > While target KT offers better-than-iid single function integration error and both root KT and target KT offer better-than-iid MMD for smooth kernels, neither the D&M (2021) root KT analysis nor our Thm. 2 target KT analysis ensures better-than-iid MMD for non-smooth kernels.  In particular, the D&M root KT algorithm is inapplicable to these kernels because they have insufficient smoothness to admit a square-root kernel, and the target KT algorithm does not reap the MMD benefits from enforcing balance in a larger, rougher RKHS.  Power KT solves these issues by leveraging a fractional power kernel that is rougher than the target but still available even when a square-root kernel does not exist.  In the uploaded revision, we have simplified our presentation of the power KT result to make it more accessible to the reader.

---

> > ### Author Response · Authors · 2021-11-21
> > **Algorithm intuition and role of $\mathbf k_{split}$**
> >
> > We agree that the paper will benefit from a review of (generalized) KT, and in the camera-ready version we will clarify the following:
> >
> > A. Generalized KT is a two phase compression procedure: in the initialization stage, KT-SPLIT partitions the input points into $2^m$ candidate coresets of size $n/2^m$ using non-uniform randomness to achieve $H_{\mathbf{k}_{{split}}}$ balance across the coresets (i.e., to ensure that integration error is small between the input point set and each candidate coreset for functions in the RKHS of $\mathbf k\_{split}$); in the refinement stage, KT-SWAP selects the candidate closest to Sin and then iteratively improves this candidate by replacing coreset points with the input points that yield the most improvement in MMD.
> >
> > B. The original (root) KT algorithm fixed $\mathbf{k}\_{{split}} =$ a square-root kernel k_rt to ensure small integration error for functions of the form $\mathbf k\_{rt}(x, .)$.  D&M (2021) then showed that small integration error for  $\mathbf k\_{rt}(x, .)$  functions in turn implied small $\mathbb{MMD}_{\mathbf k}$ (but with an added exponential dependence on dimension; please see our reply to Reviewer 5PwR).
> >
> > C. Our work (Thm. 1) shows that using $\mathbf k_{split} =$ the target kernel $\mathbf k$ instead ensures better-than-iid dimension-free integration error for each individual function in the RKHS of $\mathbf k$ and that this provides an alternative way to bound $\mathbb{MMD}_{\mathbf k}$(Thm. 2).

---

> > ### Author Response · Authors · 2021-11-21
> > **Improving presentation**
> >
> > Thank you for your helpful suggestions!  In the uploaded revision, we have reorganized the presentation and discussion of our main results, provided more intuition, and moved more minor technical details and remarks to the appendix to make the paper more accessible and less technical.  Please let us know if you see other specific opportunities for improvement!

---

### Official Review · Reviewer_5PwR · 2021-11-07

**Correctness:** 3
**Technical Novelty And Significance:** 2
**Empirical Novelty And Significance:** Not applicable
**Recommendation:** 6
**Confidence:** 3

**Main Review:**

In terms of writing, the paper is quite technical and lacks intuition. I don't think putting so many results in the main body is necessary, e.g., there are six remarks after Theorem 1.

I also have some concerns regarding the significance of the results. The summary from the authors emphasizes the error improvement from $n^{-1/4}$ to $\sqrt{\log n/ n}$, but this seems to be the main contribution of the prior work by Dwivedi and Mackey (2021). As for the current submission, the improvements are in the log factors. I didn't check the proofs of these two papers as they have 85 pages in total. Given that the prior work is twice the current length, I wonder if the authors can explain the technical novelty of the present paper that leads to the logarithmic improvement. In other words, why the prior work has the log factors?

The second contribution of the submission is  "we show that, for analytic kernels, like Gaussian and inverse multiquadric, target kernel KT admits maximum mean discrepancy (MMD) guarantees comparable to square-root KT without the need for an explicit square-root kernel." I know this is related to Fourier transforms. Do the authors know under what condition the $\mathbf{explicit}$ square-root kernels exist?

The experiments seem to be generated by the same software/scripts Dwivedi and Mackey (2021) used. It will be good if the author can also compare the run time of these algorithms. The proposed heuristics inherit the better rates the Dwivedi and Mackey (2021) algorithm provides and its quadratic time complexity. I would think that a more significant contribution would be reducing this algorithmic complexity, especially important for practical applications.

**Summary Of The Paper:**

The abstract already presents four contributions (first, second, etc.) of the paper. My summary: the submission extends the Kernel Thinning algorithm recently proposed by Dwivedi and Mackey (2021) to more general settings (kernel types). It also removes some dimension-dependent log factors in the bound.

**Summary Of The Review:**

I don't find the contribution of the current paper very significant compared to the Dwivedi and Mackey (2021) work. The paper adopts several critical theoretical contributions and algorithm designs from the prior work, maybe also the code. The writing is also technical and lacks intuitions and insights. I suggest the authors focus on improving the quadratic time-complexity instead, which should increase the significance/popularity of both works.

---

> ### Author Response · Authors · 2021-11-21
> **Author response**
>
> Thank you for the time you’ve spent reviewing our work and for your thoughtful feedback.  We address each of your concerns below.

---

> > ### Author Response · Authors · 2021-11-21
> > **New experiments and code**
> >
> > Our experiments build on the public kernel thinning code of D&M 2021.  However, due to the limitations discussed above, their experiments only used square-root kernels for the KT-SPLIT step, only involved Gaussian kernels, and only evaluated MMD (rather than exploring the downstream improvements for single-function integration error, which is what one cares about in practice).  Our new scripts provided in the supplement produce new experiments that are different than the prior work on the following fronts:
> >
> > 1. We visualize and generate generalized KT coresets with a range of different kernels beyond Gaussian, including IMQ, Laplace, and Bspline kernels. The Laplace and B-spline kernels we use do not admit square-root kernels so the root KT algorithm of D&M (2021) cannot be even run for them.
> >
> > 2. We evaluate coreset integration error on test functions both inside and outside of the RKHS including (a) moments ($f(x) = x\_1, f(x)=x\_1^2$), (b) random elements of the target kernel RKHS ($f(x) = \mathbf{k}(X', x))$, and (c) a standard numerical integration benchmark test function from the continuous integrand family.

---

> > ### Author Response · Authors · 2021-11-21
> > **Existence of square-root kernels**
> >
> > Prop. 2 of D&M (2021) provides a sufficient condition for the existence of a square-root kernel: if $\mathbf k$ is continuous and shift-invariant so that $\mathbf k(x,y) = \kappa(x-y)$ and $\kappa$ has generalized Fourier transform $\hat{\kappa}$ satisfying $\int \sqrt{\hat{\kappa}} < \infty$, then with $\mathbf{k}\_{{rt}}(x, y) = \kappa\_{{rt}}(x-y)$ for $\kappa\_{{rt}}$ the Fourier transform of $\sqrt(\hat{\kappa})$ is a valid square-root kernel.  Inversely, if $\sqrt{\hat{\kappa}} $ is not integrable, then Bochner’s theorem implies that $\kappa\_{{rt}}$ is not a kernel, making root KT inapplicable to non-smooth kernels like Laplace and non-smooth Matern.  In addition, in some cases, an exact square-root kernel exists but does not correspond to any easily computed function: this is the case for inverse multiquadric kernels.

---

> > ### Author Response · Authors · 2021-11-21
> > **Significance, technical novelty, and eliminating the exponential dimension dependence of D&M (2021)**
> >
> > We generalized the KT-SPLIT step of D&M (2021) and developed several new analyses to overcome three significant limitations in their original algorithm and analysis:
> > 1. All of the guarantees in D&M feature an exponential dependence on the dimension as summarized below which bounds single function integration error of KT-SPLIT up to constants independent of $d$:
> >
> > ||D&M (2021) guarantee|Our Thm. 1 guarantee|iid sampling|
> > |-|-|-||
> > | Compact P |$c^d \sqrt{\log(n)/n}$|$\sqrt{\log(n)/n}$|$n^{-1/4}$|
> > | Sub-Gaussian P|$c^d\log^{(d/2+1)/2}(n)/\sqrt{n}$|$\sqrt{\log(n)/n}$|$n^{-1/4}$|
> > | Subexponential P|$c^d\log^{(d+1)/2}(n)/\sqrt{n}$|$\sqrt{\log(n)/n}$|$n^{-1/4}$|
> > | Heavy-tailed P|$c^d\log n/\sqrt{n^{1-d/\rho}}$|$\sqrt{\log(n)/n}$|$n^{-1/4}$|
> >
> > As a result, in the best case, their guarantees only offer an improvement over iid sampling when the sample size $n$ is exponentially large in the dimension ($n\geq C^d$ for $C>1$), and, in the worst case (heavy tails with $\rho\leq 2d$ moments), their guarantees offer no improvement over iid sampling.
> >   - By changing their algorithm to use the target kernel $\mathbf k$ instead of a square-root kernel, we obtain the first dimension-free guarantees for KT-SPLIT: integration error $\leq \sqrt{\log n/n}$ with high probability.  An important implication is that target KT-SPLIT improves upon iid integration error even for small or moderate sample sizes and even in arbitrarily high dimensions.  Notably, the same cannot generally be said for KT-SPLIT with the square-root kernel of D&M: as we show in App. G, the corresponding integration error still grows exponentially in the dimension for the Gaussian and Matern kernels studied by D&M.
> >  - The D&M analysis identifies a deterministic relationship between target kernel MMD ($\mathbb{MMD}\_{\mathbf{k}}$) and the $L^{\infty}$ error of the square-root kernel ($\Vert (\mathbb{P}\_{{in}}-\mathbb{P}\_{{out}})\mathbf k\_{{rt}}\Vert\_{\infty}$) and then uses a chaining argument to bound the $\mathbf{k}\_{{rt}}$-$L^{\infty}$error of KT-SPLIT.  That MMD-Linf relationship (Thm. 2 in D&M) is the source of the exponential dimension dependence.  Our completely different analysis, leading to the dimension-free guarantees of Theorem 1, bounds the integration error $|\mathbb{P}\_{{in}}f-\mathbb{P}\_{{out}}f|$ of each function f in the RKHS of $\mathbf k$ directly.  Moreover, even our MMD guarantees in Theorem 2 are derived in a completely different and more direct way than those of D&M: we use a covering number argument for the unit ball of the RKHS of $\mathbf k$ combined with the individual function guarantees of Theorem 1.
> > 2. Because of the need for a square-root or square-root dominating kernel, the D&M algorithm is only applicable to smooth bounded kernels on $\mathbb{R}^d$ (i.e., bounded kernels with at least 2d continuous derivatives), a family that rules out many kernels used in practice for distribution compression including Laplace, non-smooth Matern, energy distance, and Stein kernels. In addition, their analysis based on the reduction to $\mathbf{k}\_{{rt}}$-$L^{\infty}$ error only yields improvements over iid sampling for smooth bounded kernels with rapidly decaying square-roots (ruling out the popular sinc kernel for instance).
> >   - By changing their algorithm to use the target kernel instead of a square-root kernel, we obtain better-than-iid integration error improvements for any kernel on any space (including all of the kernels listed above and even kernels on discrete or non-Euclidean spaces).
> >   - In addition, by changing their algorithm to use alpha power kernels and developing a novel power kernel analysis that interpolates between the root KT guarantees of D&M and the target KT guarantees of our Thm. 2, we also derive, for the first time, better-than-iid MMD guarantees for the non-smooth kernels ruled out by the D&M algorithm (this is the content of Thm. 3's proof in Sec. D).
> > 3. Because of their reduction from square-root kernel $L^{\infty}$ error to target kernel MMD, the D&M guarantees for root KT only improve upon iid error when $\mathbb{P}$ has sufficiently fast tail decay.  By changing their algorithm to use the target kernel, we obtain $\sqrt{\log n/n}$ integration error (a near-quadratic improvement over the $n^{-1/4}$ iid error) for every target distribution $\mathbb{P}$.
> >
> > We also agree with the reviewer that two complementary advances are needed to make the KT algorithm of D&M maximally useful:
> >
> > A. An algorithm that can accommodate any kernel encountered in practice coupled with an analysis that ensures improvement for any target distribution encountered in practice (this is what our Generalized Kernel Thinning work provides) &
> >
> > B. A scheme for speeding up such thinning algorithms to ensure that GKT can provide improved compression even for the largest input sizes (this is now an area of active work for us as we now have confidence that GKT is broadly applicable to the kernels and distributions encountered in practice).

---

> > ### Author Response · Authors · 2021-11-21
> > **Improving presentation and intuition**
> >
> > Thank you for your helpful suggestions regarding the presentation! In the uploaded revision, we have reorganized the presentation and discussion of our main results, provided more intuition, and moved more minor technical details and remarks to the appendix to make the paper more accessible and less technical. Please let us know if you see other specific opportunities for improvement!

---

### Decision · Program_Chairs · 2022-01-20

**Decision:**

Accept (Poster)

**Comment:**

The focus of the paper is kernel thinning, i.e. the extraction of a core set from a sample with good integration properties meant in MMD (maximum mean discrepancy, hence worst case) sense. Particularly, the authors propose generalizations of the kernel thinning method (Dwivedi and Mackey, 2021) which relax the assumptions imposed on the kernel (k) and the target distribution (P), and possess tighter performance guarantees.

Designing compressed representation of samples for integration is a fundamental problem in machine learning and statistics with a large number of successful applications. As assessed by the reviewers, the authors deliver important new theoretical insights in the area which can be also of clear practical interest. They also pointed out that the self-containedness of the paper could be improved and additional intuition would help the dissemination of the results among the members of the ICLR/ML audience.